# Genome-wide association study results for educational attainment aid in identifying genetic heterogeneity of schizophrenia

V. Bansal [1,2,3], M. Mitjans[1,4], C. A. P. Burik[5,6,7], R.K. Linnér [5,6,7], A. Okbay [5,7], C.A. Rietveld[6,8], M. Begemann[1,9], S. Bonn [2,3], S. Ripke[10,11,12], R. de Vlaming [5,7], M. G. Nivard [13], H. Ehrenreich[1,4] & P. D. Koellinger[5,6,7]

Higher educational attainment (EA) is negatively associated with schizophrenia (SZ). However, recent studies found a positive genetic correlation between EA and SZ. We investigate possible causes of this counterintuitive finding using genome-wide association study results for EA and SZ ($N = 443,581$) and a replication cohort (1169 controls; 1067 cases) with deeply phenotyped SZ patients. We find strong genetic dependence between EA and SZ that cannot be explained by chance, linkage disequilibrium, or assortative mating. Instead, several genes seem to have pleiotropic effects on EA and SZ, but without a clear pattern of sign concordance. Using EA as a proxy phenotype, we isolate *FOXO6* and *SLITRK1* as novel candidate genes for SZ. Our results reveal that current SZ diagnoses aggregate over at least two disease subtypes: one part resembles high intelligence and bipolar disorder (BIP), while the other part is a cognitive disorder that is independent of BIP.

[1] Clinical Neuroscience, Max Planck Institute of Experimental Medicine, Hermann-Rein-Straße 3, 37075 Göttingen, Germany. [2] Research Group for Computational Systems Biology, German Center for Neurodegenerative Diseases (DZNE), Von-Siebold-Straße 3A, 37075 Göttingen, Germany. [3] Institute of Medical Systems Biology, Center for Molecular Neurobiology, University Clinic Hamburg-Eppendorf, Falkenried 94, 20251 Hamburg, Germany. [4] DFG Research Center for Nanoscale Microscopy and Molecular Physiology of the Brain (CNMPB), Humboldtallee 23, 30703 Göttingen, Germany. [5] Complex Trait Genetics, Vrije Universiteit Amsterdam, De Boelelaan 1085 B-631, 1081 HV Amsterdam, Netherlands. [6] Institute for Behavior and Biology, Erasmus University Rotterdam, P.O. Box 1738, 3000 DR Rotterdam, Netherlands. [7] School of Business and Economics, Department of Economics, De Boelelaan 1105, 1081 HV Amsterdam, Netherlands. [8] Erasmus School of Economics, Erasmus University Rotterdam, P.O. Box 1738, 3000 DR Rotterdam, Netherlands. [9] Department of Psychiatry & Psychotherapy, University of Göttingen, Von-Siebold-Straße 5, 37075 Göttingen, Germany. [10] Analytic and Translational Genetics Unit, Massachusetts General Hospital, 02114 MA Boston, USA. [11] Stanley Center for Psychiatric Research, Broad Institute of MIT and Harvard, 02142 MA Cambridge, USA. [12] Department of Psychiatry and Psychotherapy, Charité-Universitätsmedizin Berlin, Campus Mitte, Berlin 10117, Germany. [13] Department of Biological Psychology, Vrije Universiteit Amsterdam, van der Boechorststraat 1, 1081 BT Amsterdam, Netherlands. These authors contributed equally: V. Bansal, M. Mitjans. These authors jointly supervised this work: M.G. Nivard, H. Ehrenreich, P.D. Koellinger. Correspondence and requests for materials should be addressed to P.D.K. (email: p.d.koellinger@vu.nl)

Schizophrenia (SZ) is the collective term used for a severe, highly heterogeneous and costly psychiatric disorder that is caused by environmental and genetic factors[1–4]. A genome-wide association study (GWAS) by the Psychiatric Genomics Consortium (PGC) identified 108 genomic loci that are associated with SZ[5]. These 108 loci jointly account for ≈3.4% of the variation on the liability scale for SZ[5], while all single-nucleotide polymorphisms (SNPs) that are currently measured by SNP arrays capture ≈64% (s.e. = 8%) of the variation in liability for the disease[6]. This implies that many genetic variants with small effect sizes contribute to the heritability of SZ, but most of them are unidentified as of yet. A polygenic score (PGS) based on all SNPs currently accounts for 4–15% of the variation on the liability scale for SZ[5].

Yet, this PGS does not predict any differences in symptoms or severity of the disease among SZ patients[4]. Partly, this could be because the clinical disease classification of SZ spans several different behavioural and cognitive traits that may not have identical genetic architectures. Therefore, identifying additional genetic variants and understanding through which pathways they are linked with the clinical diagnosis of SZ is an important step in understanding the aetiologies of the 'schizophrenias'[7]. However, GWAS analyses of specific SZ symptoms would require very large sample sizes to be statistically well-powered, and the currently available data sets on deeply phenotyped SZ patients are not yet large enough for this purpose.

Here, we use an alternative approach to make progress with data that is readily available—by combining GWAS for SZ and educational attainment (EA). Previous studies suggest a complex relationship between EA and SZ[8] that may be used to gain additional insights into the genetic architecture of SZ and its symptoms. In particular, phenotypic data seem to suggest a negative correlation between EA and SZ[9]. For example, SZ patients with lower EA typically show an earlier age of disease onset, higher levels of psychotic symptomatology and worsened global cognitive function[9]. In fact, EA has been suggested to be a measure of premorbid function and a predictor of outcomes in SZ. Moreover, it has been forcefully argued that retarded intellectual development, global cognitive impairment during childhood and bad school performance should be seen as core features of SZ that precede the development of psychotic symptoms and differentiate SZ from bipolar disorder (BIP)[10–14]. Furthermore, credible genetic links between SZ and impaired cognitive performance have been found[15].

In contrast to these findings, recent studies using large-scale GWAS results identified a small, but positive genetic correlation between EA and SZ ($\rho_{EA,SZ} = 0.08$)[8], and higher PGS values for SZ have been reported to be associated with creativity and greater EA[16]. Other statistically well-powered studies found that a high intelligence quotient (IQ) has protective effects against SZ[17] and reported a negative genetic correlation between IQ and SZ ($\rho_{IQ,SZ} = -0.2$)[18], suggesting the possibility that genetic effects that contribute to EA but not via IQ are responsible for the observed positive genetic correlation between SZ and EA.

Indeed, previous research by the Social Science Genetic Association Consortium (SSGAC)[8] already demonstrated that the effect of the EA-PGS on years of schooling is mediated by several individual characteristics that have imperfect or no genetic correlation with each other, including higher IQ, higher openness and higher conscientiousness. These different factors that contribute to EA seem to be related to SZ and its symptoms in complex ways[19–21]. For example, differences in openness have been reported to differentiate between patients diagnosed with SZ spectrum personality disorders (higher openness) from patients diagnosed with SZ (lower openness), while conscientiousness tends to be reduced among patients of both disorders compared to healthy controls[19].

The contributing factors to EA that have previously been identified by the SSGAC (i.e. IQ, openness and conscientiousness)[8] are phenotypically and genetically related, but by no means identical[22,23]. Specifically, the Cognitive Genomics Consortium (COGENT) reported a moderate genetic correlation between IQ and openness ($r_g = 0.48$, $P = 3.25 \times 10^{-4}$), but only a small genetic correlation of IQ and conscientiousness of 0.10 that was indistinguishable from zero ($r_g = 0.10$, $P = 0.46$)[24]. Therefore, it is appropriate to think of EA as a genetically heterogeneous trait that can be decomposed into subphenotypes that have imperfect genetic correlations with each other. If the various symptoms of SZ also have non-identical genetic architectures, this could result in a pattern where both EA and SZ share many genetic loci, but without a clear pattern of sign concordance and with seemingly contradictory phenotypic and genetic correlation results.

To explore this hypothesis and to discern it from alternative explanations, we perform a series of statistical genetic analyses using large-scale GWAS results for SZ and EA from non-overlapping samples. We start by characterizing the genetic relationship between both traits by using EA as a 'proxy phenotype'[25] for SZ. We annotate possible biological pathways, tissues and cell types implied by genetic variants that are associated with both traits and explore to what extent these variants are also enriched for association with other traits. We test if the genetic relationship between EA and SZ can be explained by chance, linkage disequilibrium (LD) or assortative mating. Furthermore, we investigate the hypothesis that the part of SZ that is different from BIP is a neurodevelopmental disorder, whereas the part of SZ that overlaps with BIP is not. Finally, we develop a formal statistical test for genetic heterogeneity of SZ using a polygenic prediction framework that leverages both the SZ and the EA GWAS results. Together, our analysis suggest that current SZ diagnoses aggregate over at least two disease subtypes: one part resembles BIP and high IQ, while the other part is a cognitive disorder that is independent of BIP.

## Results

**Genetic dependence and genetic correlation**. As a formal prelude to our study, it is conceptually important to differentiate between genetic dependence and genetic correlation. In our analyses, genetic dependence means that the genetic variants associated with EA are more likely to also be associated with SZ than expected by chance. In contrast, genetic correlation is defined by the correlation of the (true) effect sizes of genetic variants on the two traits. Thus, genetic correlation implies a linear genetic relationship between two traits whereas genetic dependence does not. Thus, two traits can be genetically dependent even if they are not genetically correlated and vice versa. One possible cause of a non-linear genetic dependence is that at least one of the traits is genetically heterogeneous in the sense that it aggregates across subphenotypes (or symptoms) with non-identical genetic architectures. Supplementary Note 1 presents a formal discussion and simulations that illustrate the data patterns that can emerge.

**Proxy-phenotype analyses**. We used the proxy-phenotype method (PPM)[25] to illustrate the genetic dependence between EA and SZ. PPM is a two-stage approach. In the first stage, a GWAS on the proxy-phenotype (EA) is conducted. The most strongly associated loci are then advanced to the second stage, which tests the association of these loci with the phenotype of interest (SZ) in an independent sample. If the two traits are genetically dependent, this two-stage approach can increase the

statistical power for detecting associations for the target trait because it limits the multiple testing burden for the phenotype of interest compared to a GWAS[8,25,26].

Our PPM analyses followed a preregistered analysis plan (https://osf.io/dnhfk/) using GWAS results on EA ($n = 363,502$)[8] and SZ (34,409 cases and 45,670 controls)[5] that were obtained from non-overlapping samples of Europeans. For replication and follow-up analyses, we used the Göttingen Research Association for Schizophrenia (GRAS) data collection[27], which has a uniquely rich and accurate set of SZ measures. The GRAS sample was not part of either GWAS.

Analyses were performed using 8,240,280 autosomal SNPs that passed quality controls in both GWAS and additional filters. We selected approximately independent lead SNPs from the EA GWAS that passed the predefined significance threshold of $P_{EA} < 10^{-5}$ and looked up their SZ results. To test if EA-associated SNPs are more strongly associated with SZ than expected by chance (referred to as 'raw enrichment' below), we conducted a Mann–Whitney test that compares the $P_{SZ}$ values of the EA-associated lead SNPs with the $P_{SZ}$ values of a set of randomly drawn, approximately LD-independent SNPs with similar minor allele frequencies (MAFs). Fig. 1 presents an overview of the proxy-phenotype analyses.

The first-stage GWAS on EA identified 506 loci that passed our predefined threshold of $P_{EA} < 10^{-5}$ (Supplementary Note 2); 108 of them were significant at the genome-wide level ($P_{EA} < 5 \times 10^{-8}$, see Supplementary Data 2). Of the 506 EA lead-SNPs, 132 are associated with SZ at nominal significance ($P_{SZ} < 0.05$), and 21 of these survive Bonferroni correction ($P_{SZ} < \frac{0.05}{506} = 9.88 \times 10^{-5}$) (Table 1). LD score regression results suggest that the vast majority of the association signal in both the EA[8] and the SZ[5] GWAS are truly genetic signals, rather than spurious signals originating from uncontrolled population stratification. Fig. 2a shows a Manhattan plot for the GWAS on EA highlighting SNPs that were also significantly associated with SZ (black crosses for $P_{SZ} < 0.05$, magenta crosses for $P_{SZ} = 9.88 \times 10^{-5}$).

A Q–Q plot of the 506 EA lead SNPs for SZ is shown in Fig. 2b. Although the observed sign concordance of 52% is not significantly different from a random pattern ($P = 0.40$), we find 3.23 times more SNPs in this set of 506 SNPs that are nominally significant for SZ than expected given the distribution of the $P$ values in the SZ GWAS results (raw enrichment $P = 6.87 \times 10^{-10}$). The observed enrichment of the 21 EA lead SNPs that pass Bonferroni correction for SZ ($P_{SZ} < \frac{0.05}{506} = 9.88 \times 10^{-5}$) is even more pronounced (27 times stronger, $P = 5.44 \times 10^{-14}$).

The effect sizes of these 21 SNPs on SZ are small, ranging from odds ratio (OR) = 1.02 (rs4500960) to OR = 1.11 (rs4378243) after correction for the statistical winner's curse[25] (Table 1). We calculated the probability that these 21 SNPs are truly associated with SZ using a heuristic Bayesian method that takes the winner's curse corrected effect sizes, statistical power and prior beliefs into account[25]. Applying a reasonable prior belief of 5%, we find that all 21 SNPs are likely or almost certain to be true positives.

**Novel SZ loci**. Of the 21 variants we identified, 12 are in LD with loci previously reported by the PGC[5] and two are in the major histocompatibility complex region on chromosome (chr) 6 and were therefore not separately reported in that study. Three of the variants we isolated (rs7610856, rs143283559 and rs28360516) were independently found in a meta-analysis of the PGC results[5] with another large-scale sample which identified 50 novel SZ SNPs[28]. Two of the 21 variants (rs756912, rs7593947) are in LD with loci recently reported in a study that also compared GWAS findings from EA and SZ using smaller samples and a less conservative statistical approach[29]. The remaining two SNPs we

identified here (rs7336518 on chr13 and rs7522116 on chr1) add to the list of empirically plausible candidate loci for SZ.

Using a proportions test, we compared the ratio of novel SNPs from ref.[28] included in our list of 132 loci that are jointly associated with EA and SZ ($P_{EA} < 10^{-5}$ and $P_{SZ} < 0.05$, yielding 6 loci) with the ratio observed in all remaining approximately independent loci with $P_{SZ} < 0.05$ in our SZ GWAS results. We found that the proportion of novel SZ SNPs is higher among the 132 loci that are informed by the EA GWAS results (Fisher's exact test $P = 2.4 \times 10^{-9}$, two-sided). Thus, using EA as a proxy-phenotype for SZ helped to predict the novel genome-wide significant findings reported in ref.[28], illustrating the power of the proxy-phenotype approach.

**Detection of shared causal loci**. The next step in our study was a series of analyses that aimed to identify reasons for the observed genetic dependence between EA and SZ and to put the findings of the PPM analysis into context. First, we probed if there is evidence that the loci identified by the PPM may tag shared causal loci for both EA and SZ (i.e. pleiotropy), rather than being in LD with different causal loci for both traits.

For each of the 21 SNPs isolated by our PPM analysis, we looked at their neighbouring SNPs within a ±500 kb window and estimated their posterior probability of being causal for EA or SZ using PAINTOR[30]. We then selected two sets of SNPs, each of which contains the smallest number of SNPs that yields a cumulative posterior probability of 90% or 50% of containing the causal locus for EA and SZ. We refer to these as broad sets (90%) and narrow sets (50%), respectively. Supplementary Note 3 and Supplementary Data 4 also contain results for the 80 and 65% credibility sets. For each of these sets, we calculated the posterior probability that it contains the causal locus for the other trait.

For the broad credibility set analyses (90%), we found 11 loci with a medium or high credibility to have direct causal effects on both EA and SZ (including one of the novel SNPs, rs7336518). Six of these loci have concordant effects on the two traits (i.e. ++ or −−) while five have discordant effects (i.e. +− or −+, Table 1). The analyses of the 50% credible sets are based on a smaller number of SNPs. This also results in lower probabilities that the SZ set contains the causal SNP for EA and vice versa. Nevertheless, our analysis with 50% credible sets show that four specific loci (rs7610856, rs320700, rs79210963 and rs7336518) had credibility of more than 15% for the other trait, providing support for the high (rs320700 and rs79210963) and medium (rs7610856 and rs7336518) credibility judgments based on the 90% sets. One of these has a discordant effect (rs7610856) while the others have a concordant effect on SZ and EA.

Overall, our analyses suggest that some of the 21 SNPs that we identified by using EA as a proxy-phenotype for SZ are likely to have direct pleiotropic effects on both traits. Of the most likely candidates for direct pleiotropic effects, three SNPs have concordant signs (rs79210963, rs7336518 and rs320700) and one has discordant signs (rs7610856).

**Biological annotations**. Biological annotation of the 132 SNPs that are jointly associated with EA ($P_{EA} < 10^{-5}$) and SZ ($P_{SZ} < 0.05$) using DEPICT (Supplementary Note 4 and Supplementary Data 5–7) points to genes that are known to be involved in neurogenesis and synapse formation (Supplementary Data 8). Some of the indicated genes, including *SEMA6D* and *CSPG5*, have been suggested to play a potential role in SZ[31,32]. For the two novel candidate SNPs reported in this study (rs7522116 and rs7336518), DEPICT points to the *FOXO6* (Forkhead Box O6) and the *SLITRK1* (SLIT and NTRK Like

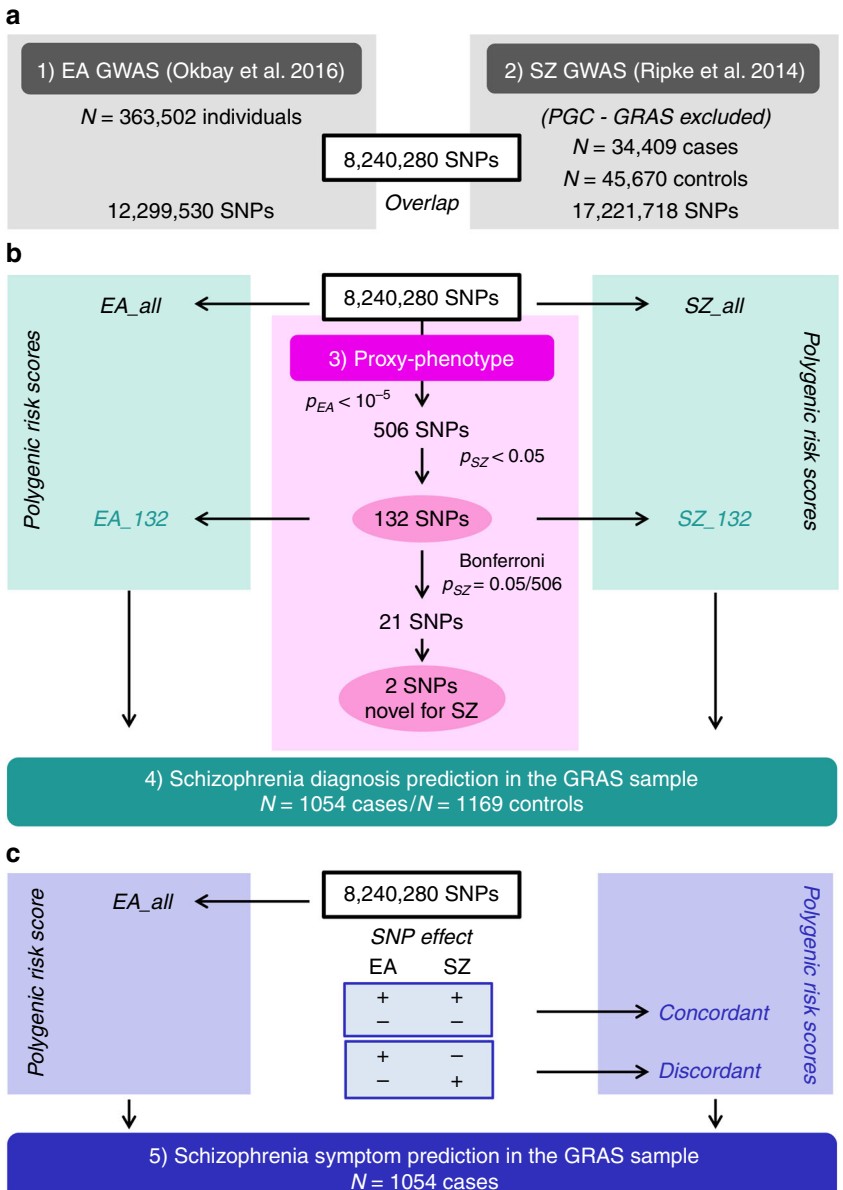

**Fig. 1** Workflow of the proxy-phenotype analyses. Notes: Educational attainment (EA) and schizophrenia (SZ) GWAS results are based on the analyses reported in refs.[5, 8]. All cohorts that were part of the SZ GWAS were excluded from the meta-analysis on EA. The GRAS data collection was not included in either the SZ or the EA meta-analysis. Proxy-phenotype analyses were conducted using 8,240,280 autosomal SNPs that passed quality control. Genetic outliers of non-European descent ($N = 13$ cases) were excluded from the analysis in the GRAS data collection

Family Member 1) genes, respectively. *FOXO6* is predominantly expressed in the hippocampus and has been suggested to be involved in memory consolidation, emotion and synaptic function[33,34]. Similarly, *SLITRK1* is also highly expressed in the brain[35], is particularly localized to excitatory synapses and promotes their development[36], and it has previously been suggested to be a candidate gene for neuropsychiatric disorders[37].

**LD-aware enrichment across different traits.** The raw enrichment *P* value reported in Fig. 2b could in principle be due to the LD structure of the EA lead SNPs that we tested. Specifically, if these EA lead SNPs have stronger LD with other SNPs in the human genome than expected by chance, this could cause the observed enrichment of this set of SNPs on SZ and other traits because higher LD increases the chance these SNPs would 'tag' causal SNPs that they are correlated with[38,39].

To assess the null hypothesis that the observed genetic dependence between EA and SZ can be entirely explained by LD patterns in the human genome, we developed an association enrichment test that corrects for the LD score of each SNP. We applied this test to the 132 SNPs that are jointly associated with EA ($P_{EA} < 10^{-5}$) and SZ ($P_{SZ} < 0.05$), i.e. the loci that were identified by using EA as a proxy-phenotype for SZ. LD scores were obtained from the HapMap 3 European reference panel[40] (Supplementary Data 9). We found significant joint LD-aware enrichment for SZ ($P = 9.57 \times 10^{-66}$), demonstrating that the genetic dependence between EA and SZ cannot be entirely explained by LD.

Furthermore, we used this test to explore if these SNPs are generally enriched for association with all (brain-related) phenotypes, or whether they exhibit some degree of outcome specificity. For this purpose, we extended the LD-aware enrichment test to 21 additional traits for which GWAS results were available in the

**Table 1 SNPs significantly associated with schizophrenia after Bonferroni correction**

| SNP-ID | EA beta | Signs concordant | SZ adj. $R^2$ (%) | SZ OR (Adj.) | EAF | Power $\alpha = 0.05/506$ (%) | Chance of direct pleiotropic effect on EA and SZ | | Posterior probability of true association with SZ prior belief ($\pi$) (%) | | | |
|---|---|---|---|---|---|---|---|---|---|---|---|---|
| | | | | | | | 90% sets | 50% sets | 0.1% | 1.0% | 5.0% | 10.0% |
| rs79210963 | −0.016 | Yes | 0.021 | 0.931 | 0.89 | 22.9 | H | M | 75.0 | 96.8 | 99.3 | 99.7 |
| rs7610856 | 0.013 | No | 0.022 | 0.955 | 0.41 | 22.8 | M | M | 74.9 | 96.8 | 99.3 | 99.7 |
| rs10896636 | 0.012 | No | 0.020 | 0.956 | 0.67 | 17.8 | H | L | 68.7 | 95.6 | 99.1 | 99.5 |
| rs756912 | −0.015 | Yes | 0.022 | 0.956 | 0.51 | 22.7 | L | L | 74.8 | 96.7 | 99.3 | 99.7 |
| rs6449503 | 0.018 | No | 0.020 | 0.961 | 0.51 | 12.9 | L | L | 60.0 | 93.7 | 98.7 | 99.3 |
| **rs7336518** | **−0.016** | **Yes** | **0.014** | **0.964** | **0.13** | **1.5** | **M** | **M** | **13.4** | **60.6** | **88.5** | **93.9** |
| rs143283559 | 0.014 | No | 0.017 | 0.965 | 0.72 | 4.6 | M | L | 32.8 | 83.0 | 96.1 | 98.0 |
| rs11210935 | 0.015 | No | 0.014 | 0.973 | 0.77 | 1.2 | L | L | 10.9 | 55.1 | 86.0 | 92.5 |
| rs77000541 | −0.014 | Yes | 0.018 | 0.974 | 0.33 | 1.6 | L | L | 14.1 | 62.2 | 89.2 | 94.3 |
| rs2819344 | 0.014 | No | 0.017 | 0.983 | 0.62 | 0.3 | H | L | 3.0 | 23.3 | 60.4 | 75.3 |
| rs4500960 | −0.013 | No | 0.017 | 1.017 | 0.47 | 0.3 | L | L | 3.0 | 23.3 | 60.4 | 75.3 |
| rs28360516 | −0.012 | No | 0.013 | 1.027 | 0.70 | 1.4 | M | L | 12.6 | 59.0 | 87.8 | 93.5 |
| **rs7522116** | **0.011** | **Yes** | **0.015** | **1.029** | **0.56** | **3.0** | **M** | **L** | **23.8** | **75.8** | **94.0** | **96.9** |
| rs7593947 | 0.014 | Yes | 0.018 | 1.040 | 0.51 | 12.5 | M | L | 59.1 | 93.5 | 98.6 | 99.3 |
| rs11694989 | 0.011 | Yes | 0.021 | 1.044 | 0.43 | 17.9 | L | L | 68.8 | 95.7 | 99.1 | 99.5 |
| rs320700 | 0.013 | Yes | 0.024 | 1.054 | 0.65 | 36.4 | H | M | 85.3 | 98.3 | 99.7 | 99.8 |
| rs3957165 | 0.015 | Yes | 0.020 | 1.056 | 0.83 | 14.7 | L | L | 63.6 | 94.6 | 98.9 | 99.4 |
| rs10791106 | 0.011 | Yes | 0.026 | 1.056 | 0.54 | 46.9 | L | L | 89.9 | 98.9 | 99.8 | 99.9 |
| rs2992632 | 0.016 | Yes | 0.025 | 1.060 | 0.74 | 36.8 | M | L | 85.5 | 98.3 | 99.7 | 99.8 |
| rs10773002 | 0.022 | Yes | 0.043 | 1.087 | 0.28 | 91.0 | L | L | 99.0 | 99.9 | 100.0 | 100.0 |
| rs4378243 | 0.019 | Yes | 0.044 | 1.112 | 0.85 | 91.5 | L | L | 99.1 | 99.9 | 100.0 | 100.0 |

Notes: The SNPs in the table are ordered by their odds ratio (OR) on schizophrenia (SZ). Effect sizes for SZ (in $R^2$ and OR) are downward adjusted for the winner's curse[25]. EA (beta) is the standardized beta of a SNP for educational attainment (EA) GWAS. $R^2$ was approximated from the winner's curse adjusted OR ratios, using the formulas described in Methods section. The winner's curse adjustment took into account that only SNPs with $P = 0.05/506$ were selected. SNPs with concordant effects on both SZ and EA are marked as 'yes' in the sign concordance column. EAF is the effect allele frequency in the SZ GWAS data. Power calculations assumed that the available GWAS sample size for SZ for each SNP consisted of 34,409 cases and 45,670 controls. The chance that a SNP has direct pleiotropic effects on EA and SZ has been evaluated with PAINTOR using sets of SNPs that have a cumulative probability of 90% or 50% to include the causal variant (see Methods and Supplementary Note 3). The posterior probability that these SNPs are truly associated with SZ was calculated using the Bayesian procedure developed by Rietveld et al.[25]. SNPs highlighted in bold are associations for SZ that have not been emphasized in the previous literature. H high, M medium, L low.

public domain (Supplementary Fig. 5 and Supplementary Data 10). Some of the traits were chosen because they are phenotypically related to SZ (e.g. neuroticism, depressive symptoms, major depressive disorder, autism and childhood IQ), while others were less obviously related to SZ (e.g. age at menarche, intracranial volume and cigarettes per day) or served as negative controls (height, birth weight, birth length and fasting (pro)insulin). The power of the LD-aware enrichment test primarily depends on the GWAS sample size of the target trait and results of our test would be expected to change as GWAS sample sizes keep growing. We found LD-aware enrichment of these SNPs for BIP, neuroticism, childhood IQ and age at menarche. However, we found no LD-aware enrichment for other brain-traits that are phenotypically related to SZ, such as depressive symptoms, subjective well-being, autism and attention deficit hyperactivity disorder. We also did not find LD-aware enrichment for most traits that are less obviously related to the brain and our negative controls. Furthermore, one of the novel SNPs we isolated shows significant LD-aware enrichment both for SZ and for BIP (rs7522116). The results suggest that the loci identified by the PPM are not simply related to all (brain) traits. Instead, they show some degree of phenotype specificity.

**Replication in the GRAS sample.** Following our preregistered analysis plan (https://osf.io/dnhfk/), we replicated the PPM analysis results in the GRAS sample (Supplementary Note 5 and Supplementary Data 11) using polygenic prediction (Supplementary Note 6, Supplementary Data 12 and Supplementary Fig. 6). The PGS (SZ_132) that is based on the 132 independent EA lead SNPs that are also nominally associated with SZ ($P_{EA} < 10^{-5}$ and $P_{SZ} < 0.05$) adds $\Delta R^2 = 7.54 - 7.01\% = 0.53\%$

predictive accuracy for SZ case–control status to a PGS (SZ_all) derived from the GWAS on SZ alone ($P = 1.7 \times 10^{-4}$, Supplementary Data 13, Model 3).

**Prediction of SZ measures among patients.** To explore the genetic architecture of specific SZ measures, we again used our replication sample (GRAS), which contains exceptionally detailed measures of SZ symptoms, severity and disease history[4,7,27]. We focused on years of education, age at prodrome, age at disease onset, premorbid IQ (approximated by a multiple-choice vocabulary test), global assessment of functioning (GAF), the clinical global impression of severity (CGI-S) as well as positive and negative symptoms (PANSS positive and negative, respectively) among SZ patients (N ranges from 903 to 1039, see Supplementary Note 5). Consistent with the idea that EA is a predictor of SZ measures, our phenotypic correlations show that higher education is associated with later age at prodrome, later onset of disease and less severe disease symptoms among SZ patients (Supplementary Note 7, Supplementary Data 15 and Supplementary Fig. 7).

Our most direct test for genetic heterogeneity of SZ is based on PGS analyses that we performed using the detailed SZ measures among GRAS patients. If SZ is genetically heterogeneous, there is potentially relevant information in the sign concordance of individual SNPs with EA traits that may improve the prediction of symptoms (Supplementary Note 1). We use a simple method to do this here: first, we construct a PGS for SZ that contains one SNP per LD-block that is most strongly associated with SZ. Overall, this score (SZ_all) contains 349,357 approximately LD-independent SNPs. Next, we split SZ_all into two scores, based on sign-concordance of the SNPs with SZ and EA. More specifically,

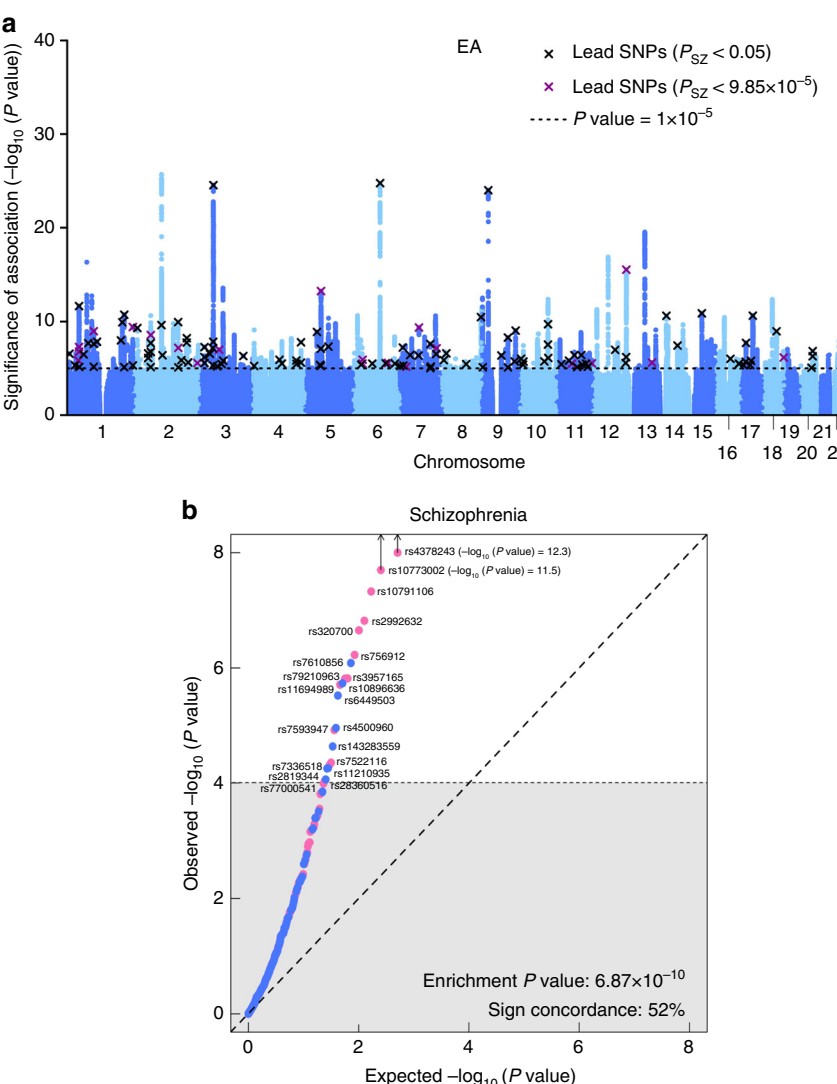

**Fig. 2** Results of the proxy-phenotype analyses. Notes: **a** Manhattan plot for educational attainment (EA) associations ($n = 363,502$). The x axis is the chromosomal position, and the y axis is the significance on a $-\log_{10}$ scale (two-sided). The black dashed line shows the suggestive significance level of $10^{-5}$ that we specified in our preregistered analysis plan. Black and magenta crosses identify EA-associated lead-SNPs that are also associated with SZ at nominal or Bonferroni-adjusted significance levels, respectively. **b** Q–Q plot of the 506 EA-associated SNPs for schizophrenia (SZ) ($n = 34,409$ cases and $n = 45,670$ controls). SNPs with concordant effects on both phenotypes are pink, and SNPs with discordant effects are blue. SNPs outside the grey area (21 SNPs) pass the Bonferroni-corrected significance threshold that corrects for the total number of SNPs we tested ($P < 0.05/506 = 9.88 \times 10^{-5}$) and are labelled with their rs numbers. Observed and expected P values are on a $-\log_{10}$ scale. For the sign concordance test: $P = 0.40$, two-sided

one score contains all estimated SZ effects of SNPs that have concordant signs for both traits (174,734 SNPs with $++$ or $--$ on both traits, *Concordant*) while the other contains the estimated SZ effects of the remaining SNPs with discordant effects (174,623 SNPs with $+-$ or $-+$, *Discordant*). Note that splitting the *SZ_all* score this way is not expected to improve the prediction of symptoms if they share the same genetic architecture (i.e. if SZ was a genetically homogenous trait). We test this null hypothesis with an *F* test that compares the predictive performance of models that include (i) the *SZ_all* and the EA score (*EA_all*) and (ii) the *Concordant*, *Discordant*, and *EA_all* scores (Supplementary Note 1). We also compare the performance of both of these models to a baseline that only includes the *SZ_all* score as a relevant predictor.

We found that the *EA_all* PGS is associated with years of education ($P = 1.0 \times 10^{-6}$) and premorbid IQ ($P = 2.7 \times 10^{-4}$) among SZ patients (Table 2). Consistent with earlier results[4], we also found that none of the SZ measures can be predicted by the

PGS for SZ (*SZ_all*, Table 2). However, splitting the PGS for SZ based on the sign-concordance of SNPs with EA (*Concordant* and *Discordant*) increased predictive accuracy significantly for severity of disease (GAF ($p_F = 0.023$)) and symptoms (PANSS negative ($p_F = 0.007$)) (Table 2). This increase in predictive accuracy is evidence for genetic heterogeneity of SZ (Supplementary Note 1). Specifically, our results indicate that SZ patients with a high genetic propensity for EA have better GAFs and less severe negative symptoms (PANSS negative). However, if the high genetic predisposition for EA is primarily due to loci that also increase the risk for SZ (i.e. high values on the *Concordant* score), this protective effect is attenuated. We repeated these analyses excluding patients who were diagnosed with schizoaffective disorder (SD, $N = 198$) and found similar results, implying that our findings are not only due to the presence of patients with SD (Supplementary Note 8, Supplementary Data 18).

We note that this implementation of our test for heterogeneity of SZ (Supplementary Note 1) is based on a conservative pruning

**Table 2 Polygenic prediction of schizophrenia measures in the GRAS patient sample**

| | | Years of education[a] | Age at prodrome | Age at disease onset | Premorbid IQ[a] | GAF[b] | CGI-S[b] | PANSS positive[b] | PANSS negative[b] |
|---|---|---|---|---|---|---|---|---|---|
| **Baseline model** | | | | | | | | | |
| SZ_all | Stand. beta | 0.001 | −0.041 | −0.056 | −0.063 | −0.024 | 0.041 | 0.033 | 0.043 |
| | P value | 0.976 | 0.297 | 0.129 | 0.090 | 0.510 | 0.249 | 0.364 | 0.253 |
| EA_all | Stand. beta | 0.182** | 0.005 | −0.002 | 0.149** | 0.068* | −0.057 | 0.001 | −0.051 |
| | P value | $4.4 \times 10^{-09}$ | 0.884 | 0.961 | $7.2 \times 10^{-6}$ | 0.029 | 0.065 | 0.981 | 0.107 |
| | Adj. $R^2$ | 0.0612 | 0.0023 | 0.0047 | 0.0417 | 0.0655 | 0.0816 | 0.0711 | 0.0243 |
| | ΔAdj. $R^2$[c] | 0.0312 | −0.0010 | −0.0009 | 0.0209 | 0.0035 | 0.0023 | −0.0010 | 0.0015 |
| **Split model** | | | | | | | | | |
| Concordant | Stand. beta | −0.013 | −0.019 | −0.031 | −0.043 | −0.096* | 0.050 | 0.079 | 0.125** |
| | P value | 0.751 | 0.665 | 0.456 | 0.326 | 0.022 | 0.232 | 0.059 | 0.0036 |
| Discordant | Stand. beta | 0.014 | −0.030 | −0.035 | −0.034 | 0.066 | <0.001 | −0.039 | −0.072 |
| | P value | 0.730 | 0.515 | 0.409 | 0.437 | 0.112 | 0.996 | 0.351 | 0.090 |
| EA_all | Stand. beta | 0.191** | 0.002 | −0.002 | 0.153** | 0.122** | −0.074 | −0.039 | −0.118** |
| | P value | $1.0 \times 10^{-06}$ | 0.965 | 0.953 | $2.7 \times 10^{-4}$ | 0.002 | 0.058 | 0.319 | 0.003 |
| | Adj. $R^2$ | 0.0604 | 0.0012 | 0.0037 | 0.0406 | 0.0694 | 0.0811 | 0.0728 | 0.0306 |
| | ΔAdj. $R^2$[c] | 0.0304 | −0.0021 | −0.0019 | 0.0198 | 0.0074 | 0.0018 | 0.0007 | 0.0078 |
| n | | 1039 | 915 | 1043 | 903 | 1010 | 1014 | 1009 | 1002 |
| $\Delta R^2$ (Split−baseline model) | | −0.0008 | −0.0011 | −0.0010 | −0.0011 | 0.0039 | −0.0005 | 0.0017 | 0.0063 |
| P value from F test[d] | | 0.698 | 0.907 | 0.968 | 0.891 | 0.023* | 0.479 | 0.098 | 0.007** |

*Notes*: Linear regression using the first ten genetic principal components as control variables [a]Age of onset was included as covariate [b]Medication was included as covariate [c]Change in *Adj. R²* of the models compared to a model that only contains the *SZ_all* score and the control variables [d]P value from F test refers to improvement in split model compared to baseline model *Significance at $P < 0.05$ **Significance at $P < 0.01$ Stand standardized.

algorithm that controls for LD both within and across the *Concordant* and *Discordant* scores. This limits the number of genetic markers in both of these scores, their expected predictive accuracy and the power of the test. As an alternative, we also used a less conservative approach that only prunes for LD within scores, yielding 260,441 concordant and 261,062 discordant SNPs. Split scores based on this extended set of SNPs have higher predictive accuracy for all the SZ measures that we analysed (Supplementary Data 22), reaching $\Delta R^2 = 1.12\%$ ($p_F = 0.0004$) for PANSS negative.

Finally, we show that randomly splitting the *SZ_all* score does not yield any gains in predictive accuracy (Supplementary Data 19).

**Genetic differences between SZ and bipolar.** The ongoing debate about what constitutes the difference between SZ and BIP[10–14] suggests an additional possibility to test for genetic heterogeneity among SZ cases. While SZ and BIP share psychotic symptoms such as hallucinations and delusions, scholars have argued that SZ should be perceived as a neurodevelopmental disorder in which cognitive deficits precede the development of psychotic symptoms, while this is not the case for BIP[10–14]. However, cognitive deficits during adolescence are currently not a diagnostic criterion that formally differentiates SZ from BIP. As a result, many patients who are formally diagnosed with SZ did not suffer from cognitive impairments in their adolescent years, but their disease aetiology may be different from those who do. These differences in disease aetiology may be visible in how the non-shared part of the genetic architecture of SZ and BIP is related to measures of cognition, such as EA and childhood IQ.

We tested this by using genome-wide inferred statistics (GWIS)[41] to obtain GWAS regression coefficients and standard errors for SZ that are 'purged' of their genetic correlation with BIP and vice versa (yielding 'unique' SZ$_{(min BIP)}$ and 'unique' BIP$_{(min SZ)}$ results, respectively). We repeated the look-up of the EA-associated lead SNPs in those summary statistics and find that the enrichment is weaker than in the SZ GWAS results that did not control for genetic overlap between SZ and BP (Supplementary Note 9).

We then computed genetic correlations of these GWIS results with EA, childhood IQ and (as a non-cognitive control trait) neuroticism using bivariate LD score regression[42], and compared the results to those obtained using ordinary SZ and BIP GWAS results (Supplementary Note 10).

In line with earlier findings[8,42], we see a positive genetic correlation of ordinary SZ and BIP with EA. However, the genetic correlations between 'unique' SZ$_{(min BIP)}$ with EA and childhood IQ are negative and significant ($r_g = -0.16$, $P = 3.88 \times 10^{-4}$ and $r_g = -0.31$, $P = 6.00 \times 10^{-3}$, respectively), while the genetic correlations of 'unique' BIP$_{(min SZ)}$ with EA and IQ remain positive ($r_g \approx 0.3$) (Fig. 3, Supplementary Data 24). Thus, the slightly positive genetic correlation between SZ and EA[8,42] can be entirely attributed to the genetic overlap between SZ and BIP[41], a result recently replicated using genomic structural equation modelling[43]. Overall, these results add to the impression that current clinical diagnoses of SZ aggregate over various non-identical disease aetiologies.

**Simulating assortative mating.** Finally, simulations show that assortative mating is unlikely to be a major cause of the observed level of genetic dependence between EA and SZ (Supplementary Note 11, Supplementary Fig. 9).

## Discussion
We explored the genetic relationship between EA and SZ using large, non-overlapping GWAS samples. Our results show that EA-associated SNPs are much more likely to be associated with SZ than expected by chance, i.e. both traits are genetically dependent. Overall, we isolated 21 genetic loci that are credibly associated with SZ by using EA as a proxy-phenotype, including two novel candidate genes, *FOXO6* and *SLITRK1*. Furthermore, we showed that EA GWAS results help to predict future GWAS findings for SZ in even larger samples.

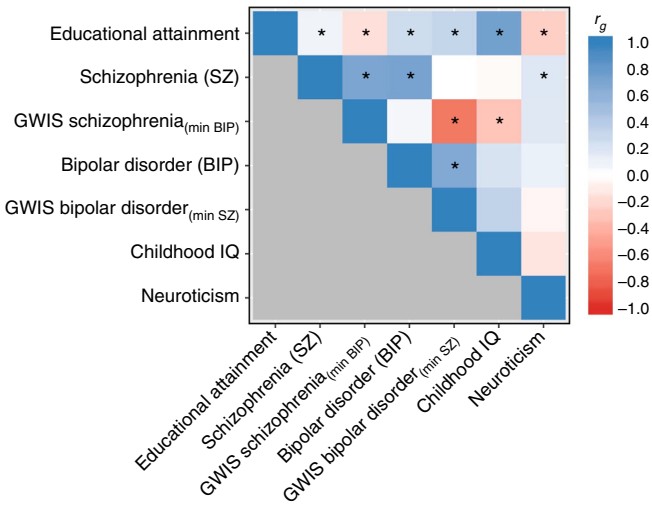

**Fig. 3** Genetic correlations of GWAS and GWIS results. Notes: The heatmap displays the genetic correlations across seven sets of GWAS or GWIS summary statistics. Genetic correlations were estimated with LD score regression [42]. The colour scale represents the genetic correlations ranging from –1 (red) to 1 (blue). Asterisks denote significant genetic correlations at $P$ value < 0.01

Biological annotation of a broader set of SNPs that are jointly associated with EA ($P_{EA} < 10^{-5}$) and SZ ($P_{SZ} < 0.05$) points to neurogenesis and synapse formation as potentially important pathways that may influence both traits.

However, the genetic loci that are associated with both traits do not follow a systematic sign pattern that would correspond to a strong positive or negative genetic correlation. Our follow-up analyses demonstrated that this pattern of strong genetic dependence but weak genetic correlation between EA and SZ cannot be fully explained by LD or assortative mating.

Instead, our results are most consistent with the idea that EA and SZ are both genetically heterogeneous traits that aggregate over various subphenotypes or symptoms with non-identical genetic architectures. Specifically, our results suggest that current SZ diagnoses aggregate over at least two disease subtypes: one part resembles BIP and high IQ (possibly associated with *Concordant* SNPs), where better cognition may also be genetically linked to other BIP features such as higher energy and drive, while the other part is a cognitive disorder that is independent of BIP (possibly influenced by *Discordant* SNPs). This latter subtype bears similarity with Kraepelin's description of dementia praecox[11]. Overall, our pattern of results resonates with the idea that cognitive deficits in early life may be an important differentiating factor between patients with BIP versus SZ psychosis.

Moreover, splitting the PGS for SZ into two scores based on the sign concordance of SNPs with EA enables the prediction of disease symptoms and severity from genetic data for the first time to some extent. We showed that this result is not driven by patients with SD and it cannot be repeated by randomly splitting the SZ score. Obviously, further replication of our results in other samples with high-quality SZ measures would be highly desirable.

The many sign-concordant loci that increase the risk for SZ but also improve the chance for higher education point to possible side-effects of pharmacological interventions that may aim to target biological pathways that are implicated by pleiotropic loci. Indeed, exploring pleiotropic patterns of disease-associated genes across a broad range of phenotypes (including social-scientific ones such as EA or subjective well-being[26]) may be a viable

strategy to identify possible side-effects of new pharmacological products at early stages of drug development in the future.

Although the complexity of SZ remains astonishing, our study contributes to unravelling this complexity by starting at a genetic level of analysis using well-powered GWAS results. Our results provide some hope that a psychiatric nosology that is based on biological causes rather than pure phenotypical classifications may be feasible in the future. Studies that combine well-powered GWASs of several diseases and from phenotypes that represent variation in the normal range such as EA are likely to play an important part in this development. However, deep phenotyping of large patient samples will be necessary to link GWAS results from complex outcomes such as EA and SZ to specific biological disease subgroups.

## Methods

**GWAS.** The principal investigators of all cohorts obtained informed consent from all study participants and approval from Institutional Review Boards (IRB) at their respective institution. We obtained GWAS summary statistics on EA from the SSGAC. The results are based on Okbay et al.[8], including the UK Biobank. The PGC shared GWAS summary statistics on SZ with us that were reported in Ripke et al.[5], but excluded data from our replication sample (GRAS), yielding a total sample size of $n = 34,409$ cases and $n = 45,670$ controls.

All cohorts that were part of both studies[5,8] were excluded from the meta-analysis on EA, yielding non-overlapping GWAS samples and $n_{EA} = 363,502$. The original EA results file contained 12,299,530 genetic markers, compared to 17,221,718 in the SZ results file.

We applied the following additional quality control steps:

1. To maximize statistical power, we excluded SNPs that were missing in large parts of the two samples. Specifically, we continued with SNPs that were available in at least 19 out of 50 cohorts in the SZ results[5] (the actual N per SNP was not provided in the SZ GWAS summary statistics) and in $N > 200,000$ in the EA meta-analysis[8]. This step excluded 3,778,914 and 6,369,138 genetic markers for EA and SZ, respectively.
2. We dropped SNPs that were not available in both GWAS results files. This step restricted our analyses to the set of available genetic markers that passed the quality-control filters in both the EA and the SZ GWAS results, leaving us with 8,403,560 autosomal SNPs.
3. We dropped six SNPs with non-standard alleles (i.e. not A, C, T or G) and two SNPs with mismatched effective alleles. Furthermore, we dropped 163,272 SNPs in the first and the 99th percentile of the distribution of differences in MAF in the two results files. This final step eliminated SNPs that were likely to be affected by coding errors, strand flips or substantial differences in MAF in the EA and SZ samples.

The remaining 8,240,280 autosomal SNPs were used in the proxy-phenotype and prediction analyses.

**Proxy-phenotype method.** Look-up: We conducted our proxy-phenotype analyses following a pre-registered analysis plan (https://osf.io/dnhfk/), using the 8,240,280 autosomal SNPs that passed quality control. We selected $10^{-5}$ as the default $P$ value threshold to identify EA-associated SNPs prior to carrying out the proxy-phenotype analyses (Supplementary Note 2).

To select approximately independent SNPs from the EA GWAS results, we applied the clumping procedure in PLINK version 1.9[44,45] using $r^2 > 0.1$ and 1,000,000 kb as the clumping parameters and the 1000 Genomes phase 1 version 3 European reference panel[46] to estimate LD among SNPs. This algorithm assigns the SNP with the smallest $P$ value as the lead SNP in its 'clump'. All SNPs in the vicinity of 1,000,000 kb around the lead SNP that are correlated with it at $r^2 > 0.1$ are assigned to this clump. The next clump is formed around the SNP with the next smallest $P$ value, consisting of SNPs that have not been already assigned to the first clump. This process is iterated until no SNPs remain with $P < 10^{-5}$, leading to 506 approximately independent EA-associated lead SNPs. 108 of the 506 EA-associated lead SNPs are genome-wide significant ($P < 5 \times 10^{-8}$).

We looked up the SZ GWAS results for these 506 EA-associated lead SNPs. Results for all 506 SNPs are reported in Supplementary Data 2 and Fig. 2.

In order to investigate the novelty of the findings, we extracted all the SNPs in LD with these 21 SNPs at $r^2 \geq 0.1$ with a maximum distance of 1000 kb using the 1000 Genomes phase 1 European reference panel.

Bayesian credibility of results: We probed the credibility of our proxy-phenotype association results using a heuristic Bayesian calculation following Rietveld et al. (Supplementary Information pp. 13–15)[25]. We focus on the 21 EA-associated lead SNPs that are also associated with SZ after Bonferroni correction.

Bayes' rule implies that the probability that an association is true given that we observe significance is given by

$$P\left(H_1|t>t_{\alpha/2}\right) = \frac{P\left(t>t_{\alpha/2}|H_1\right)P(H_1)}{P\left(t>t_{\alpha/2}|H_1\right)P(H_1)+P\left(t>t_{\alpha/2}|H_0\right)P(H_0)}$$
$$= \frac{(\text{power})(\pi)}{(\text{power})(\pi)+(\alpha)(1-\pi)}$$

'Power', as well as the significance test, are two-sided, $\pi$ is the prior belief that the SNP is truly associated and $\alpha$ is the significance threshold used for testing (in our case, $\alpha = \frac{0.05}{506} = 9.88 \times 10^{-5}$).

To calculate power for each SNP, we computed the winner's curse corrected OR using the procedure described in Rietveld et al. (Supplementary Information pp. 7–13)[47] for the $\alpha$ threshold of $9.88 \times 10^{-5}$. Because the actual sample size per SNP is not reported in the SZ GWAS summary statistics, we furthermore assumed that each SNP was available in the entire sample of 34,409 cases and 45,670 controls (i.e. the PGC results from Ripke et al.[5] excluding the GRAS data collection).

An important question is which prior beliefs are reasonable starting points for these Bayesian calculations. For an arbitrarily chosen SNP, the most conservative reasonable prior would assume that each truly associated SNP has the same effect size as the strongest effect size that was actually observed in the data. If one divides the SNP-based heritability of the trait by that effect size in $R^2$ units, one obtains a lower bound for the number of SNPs that can be assumed to be truly associated. To aid this line of thinking, we converted the winner's curse corrected OR of our 21 SNPs into $R^2$ using

$$R^2 = \left(\frac{d}{\sqrt{d^2+a}}\right)^2$$

where $d$ is Cohen's $d$, which is calculated as

$$d = \ln(\text{Odds})\frac{\sqrt{3}}{\pi}$$

and $a$ is a correction factor that adjusts for the MAF of the SNP. This correction factor is calculated as

$$a = \frac{(n_1+n_2)^2}{n_1 n_2}$$

where $n_1 = N \times \text{MAF}$ and $n_2 = N \times (1-\text{MAF})$[48].

The largest effect size in $R^2$ that we observe in our results is rs4378243 with 0.044%. The SNP-based heritability of SZ is ≈21%[49]. Thus, if all causal SZ SNPs would have an effect of $R^2 = 0.044\%$, we would expect that ≈500 truly causal loci exist. The chance of finding any one of them by chance from a set of ≈500,000 independent loci in the human genome is ≈0.1%. (Our pruning algorithm of SNPs that passed QC leads to only 223,065 independent loci. Thus, assuming 500,000 independent loci in these calculations is conservative.) However, in reality most truly associated loci for SZ will surely have smaller effects than that. Thus, a prior belief of ≈0.1% is certainly too conservative.

Furthermore, the SNPs we investigate are not arbitrary but selected based on their association with another, genetically related cognitive trait (EA) in a very large, independent sample. Thus, a prior belief of 1 or 5% that these SNPs are also associated with SZ is probably more reasonable. As an upper bound, we assume that 10% of all loci are causal. Thus, the chance to pick any one of them by chance would be 10%.

Table 1 displays the winner's curse corrected effect size of the 21 EA-associated lead SNPs that are also associated with SZ after Bonferroni correction. It also shows the posterior probability that these SNPs are truly associated with SZ given our results for prior beliefs ranging from 0.1, 1, 5 to 10%. Thirteen of these SNPs have posterior probabilities of being true positives of >50% for even the most conservative prior. For a more realistic prior belief of 5%, all 21 SNPs are likely or almost certain to be true positives.

Sign concordance: We compared the signs of the beta coefficients of the 506 EA lead SNPs ($P_{\text{EA}} < 10^{-5}$) with the beta coefficients for SZ. If the signs were aligned, we assigned a '1' to the SNP and '0' otherwise. By chance, sign concordance is expected to be 50%. We tested if the observed sign concordance is different from 50% using the binomial probability test[50]. 263 of the 506 SNPs have the same sign (52%, $P = 0.40$, two-sided).

Sign concordance is 58% ($P = 0.10$, two-sided) in the set of 132 EA lead SNPs that are also nominally significant for SZ ($P_{\text{EA}} < 1 \times 10^{-5}$ and $P_{\text{SZ}} < 0.05$).

Finally, for the 21 SNPs that passed Bonferroni correction for SZ ($P_{\text{EA}} < 1 \times 10^{-5}$ and $P_{\text{SZ}} < 9.88 \times 10^{-5}$), sign concordance is 62% ($P = 0.38$, twosided).

Raw enrichment factor (not corrected for LD score of SNPs): Because EA and SZ are highly polygenic, we tested for enrichment by taking the actual distribution of $P$ values in the GWAS result files into account.

Due to the polygenic architecture of both traits, it is expected to find some EA-associated SNPs that are also associated with SZ just by chance even if both traits are genetically independent. Under this null hypothesis, the expected number of

EA-associated lead SNPs that are also significantly associated with SZ is

$$E_{H_0}\left[N_{\text{S,EA}\rightarrow\text{SZ}}\right] = N_{T,\text{EA}} \times \tau_{P_{\text{EA}}} \times \tau_{P_{\text{SZ}}}$$

where $N_{T,\text{EA}}$ is the total number of independent lead SNPs in the EA GWAS results, and $\tau_{P_{\text{EA}}}$ and $\tau_{P_{\text{SZ}}}$ are the shares of SNPs in $N_{T,\text{EA}}$ that have $P$ values for EA and SZ below a certain threshold, respectively.

We define the raw enrichment factor as

$$N_{\text{S,EA}\rightarrow\text{SZ}}/E\left[N_{\text{S,EA}\rightarrow\text{SZ}}\right]$$

where $N_{\text{S,EA}\rightarrow\text{SZ}}$ is the observed independent number of SNPs that pass both the $P$ value thresholds $P_{\text{EA}}$ and $P_{\text{SZ}}$.

We obtained $N_{T,\text{EA}}$ by applying the clumping procedure described above (PPM) without a $P$ value threshold for EA, leading to 222,289 independent EA lead SNPs in our merged GWAS results file. For $P_{\text{EA}} < 10^{-5}$, we found 506 SNP ($\tau_{P_{\text{EA}}} = \frac{506}{222,289} = 0.2276\%$).

The Bonferroni threshold for testing 506 independent hypothesis is $P_{\text{SZ}} < \frac{0.05}{506} = 9.88 \times 10^{-5}$. There are 341 independent SNPs in the SZ results that pass this threshold, thus $\tau_{P_{\text{SZ}}} = \frac{341}{222,289} = 0.1534\%$. Therefore, we expect $[N_{\text{S,EA}\rightarrow\text{SZ}}] = 222,289 \times 0.2276\% \times 0.1534\% = 0.776$ (i.e. less than one) SNP to be jointly associated with both traits under the hull hypothesis of no genetic overlap. At these $P$ value thresholds, we actually observe $N_{\text{S,EA}\rightarrow\text{SZ}} = 21$ SNPs, implying a raw enrichment factor of $\frac{21}{0.776} = 27$.

For $P_{\text{SZ}} < 0.05$, we found 17,935 SNP ($\tau_{P_{\text{SZ}}} = \frac{17,935}{222,289} = 8.068\%$). Thus, $[N_{\text{S,EA}\rightarrow\text{SZ}}] = 222,289 \times 0.2276\% \times 8.068\% = 41$. At this more liberal $P$ value threshold, we actually observe $N_{\text{S,EA}\rightarrow\text{SZ}} = 132$ SNPs, implying a raw enrichment factor of $\frac{132}{41} = 3.23$.

Raw enrichment P value (not corrected for LD score of SNPs): Following Okbay et al.[26], we performed a non-parametric test of joint enrichment that probes whether the EA lead SNPs are more strongly associated with SZ than randomly chosen sets of SNPs with MAF within one percentage point of the lead SNP. To perform our test, we randomly drew ten matched SNPs for each of the 506 EA lead SNPs with $P_{\text{EA}} < 10^{-5}$.

We then ranked the $506 \times 10$ randomly matched SNPs and the original 506 lead EA SNPs by $P$ value and conducted a Mann–Whitney test[51] of the null hypothesis that the $P$ value distribution of the 506 EA lead SNPs are drawn from the same distribution as the $506 \times 10$ randomly matched SNPs. We reject the null hypothesis with $P = 6.872 \times 10^{-10}$ ($Z = 6.169$, two-sided). As a negative control test, we also calculated the raw enrichment $P$ value of the first randomly drawn, MAF-matched set of SNPs against the remaining nine sets, yielding $P = 0.17$.

Repeating this raw enrichment test for the subset of 21 EA-associated SNPs that remained significantly associated with SZ after Bonferroni correction (threshold $P_{\text{SZ}} < \frac{0.05}{506} = 9.88 \times 10^{-5}$) yields $P = 5.44 \times 10^{-14}$ ($Z = 7.521$, two-sided). The negative control test based on the raw enrichment $P$ value of the first randomly drawn, MAF-matched set of SNPs against the remaining nine sets yields $P = 0.34$.

**GWAS catalogue look-up.** In order to investigate the novelty of the 21 SNP associations that were found significant for SZ after Bonferroni correction, reported in Table 1, we performed a look-up in the GWAS catalogue[52] (revision 2016-08-25, downloaded on 2016-08-29, https://www.ebi.ac.uk/gwas/api/search/downloads/full) with the SNPs and all their 'LD partners' (i.e. all SNPs with an $r^2 > 0.5$ within a 250 kb window). The LD partners were extracted with PLINK[44] using a version of the 1000G reference panel specifically harmonized to combine 1000G phase 1 and phase 3 imputed data[53], and the reference panel has been described previously[26]. The result of the GWAS catalogue look-up is reported in Supplementary Data 3.

**Prediction of future GWAS loci for SZ.** To identify LD partners and to clump our GWAS results, we used a threshold of $r^2 > 0.1$ and a 1,000,000 kb window in the 1000 Genomes phase 1 version 3 European reference panel. Our SZ summary statistics contained 51,721 approximately independent SNPs with $P_{\text{SZ}} < 0.05$. We identified 21,430 SNPs in LD with the 50 novel SNPs reported in ref.[28] and 54,425 SNPs in LD with the 128 genome-wide significant loci that were previously reported[5]. We removed SNPs in LD with the previously GWAS hits from our analyses because those SNPs could (by definition) not be identified as novel. The remaining set of 51,528 approximately independent SNPs with $P_{\text{SZ}} < 0.05$ in our SZ GWAS results contained one proxy for each of the 50 novel SNPs in ref.[28]. After removing SNPs in LD with previous GWAS hits, 110 SNPs with $P_{\text{SZ}} < 0.05$ also exhibited $P_{\text{EA}} < 10^{-5}$ in the independent EA GWAS sample. Of those 110 SNPs, six were identified as novel SZ loci in the most recent GWAS dataset expansion[28]. Using Fisher's exact test, we rejected the null hypothesis that the proportion of novel SNPs (6/110 vs 50/51528) is equal in the two sets ($P = 2.4 \times 10^{-9}$, two-sided). Furthermore, as a robustness check, we performed the analysis again by excluding the SNPs with MAF ≤ 0.1 and found similar results ($P = 1.2 \times 10^{-6}$). Thus, we conclude that conditioning GWAS results on SZ with independent GWAS evidence on EA significantly outperforms pure chance in predicting GWAS results on SZ from even larger samples.

**Pleiotropy between EA and SZ**. To explore if the loci identified by our PPM may have direct pleiotropic effects on EA and SZ, we applied genetic fine mapping using full GWAS results for both traits. Our procedure was as follows:

First, the SZ and EA GWAS results were merged into a single file and aligned such that the reference allele is identical. Ambiguous SNPs or SNPs which may be subject to strands flips were removed. Second, all SNPs within 500 kb upstream and downstream of the 21 significant lead SNPs from the PPM analyses were extracted. The pairwise LD between all SNPs in each window was computed. We then ran PAINTOR 3.0[30] which estimates the posterior probability of any SNP within a locus to be causal. We applied this procedure for EA and SZ. We then selected a 90% credibility set for EA and SZ, which reflects the broadest possible set of SNPs whose posterior probability covers 90% of the total posterior probability at that locus. We predetermined the maximum number of true causal loci to be 2. For the EA 90% credibility set, we then determined the posterior probability that this set contains the causal locus for SZ and vice versa. As the size of this set fluctuates between <1% of the locus size to ~30% of the total locus size (i.e. a more narrow set can be identified for some loci compared to others), we also computed 80, 65 and 50% credibility sets for EA and SZ (Supplementary Note 3 and Supplementary Data 4), which all have increasingly narrower sets of SNPs (the 50% credibility set is the narrowest set we investigated). Finally, we compute the ratio of the cross trait credibility for the 90% sets and the proportion of SNPs in the locus, which reflects the enrichment of signal over the baseline where each SNP is equally credible. We classify the probability of a locus being pleiotropic as low, medium, or high if the posterior probability of both the EA set on SZ and the SZ set on EA are <15, 15–45 or >45% respectively.

**Biological annotations**. To gain insights into possible biological pathways that are indicated by the PPM results, we applied DEPICT[8,54] using a false discovery rate threshold of ≤0.05. DEPICT is a data-driven integrative method that uses reconstituted gene sets based on massive numbers of experiments measuring gene expression to (1) prioritize genes and gene sets and (2) identify tissues and cell types wereprioritised genes are highly expressed. The input for our analyses (DEPICT version 1 release 194) were the 132 EA lead SNPs that are also nominally associated with SZ.

For these 132 EA lead SNPs, we also used DEPICT to determine the enrichment of expression in particular tissues and cell types by testing whether the genes overlapping the GWAS loci are highly expressed in any of 209 Medical Subject Heading (MeSH) annotations.

To identify independent biological groupings, we computed the pairwise Pearson correlations of all significant gene sets using the 'network_plot.py' script provided with DEPICT. Next, we used the Affinity Propagation method on the Pearson distance matrix for clustering[55]. The Affinity Propagation method automatically chooses an exemplar for each cluster.

Furthermore, we prioritize genes using DEPICT. Any particular locus centred on a SNP may contain multiple genes. One straightforward approach is to nominate a gene that is closest to the SNP. But this approach does not consider if the expression of the gene is likely to be altered or regulated by the causal site in the locus. Therefore, we used DEPICT to map genes to associated loci, which prioritize important genes that share similar annotations in bioinformatic databases.

Significant reconstituted gene sets, tissues, cell types and prioritized genes identified by DEPICT are described in Supplementary Note 4.

**LD-aware enrichment of PPM results across different traits**. For SNP $i$ in trait $j$, the expected $\chi^2$ statistic can be calculated as

$$E\left[Z_{ij}^2\right] = \left(N_j \times h_j^2 \times \text{LDscore}_i/M\right) + (1 + \text{Na})_j$$

where $N$ is the sample size of the target trait $j$, $h^2$ is the heritability of trait $j$, $\text{LDscore}_i = \sum_k r_{ik}^2$ for SNP $i$ is calculated using HapMap3 SNPs from European-ancestry, $M$ is the number of SNPs included in the calculation of the LD score ($n = 1,173,569$ SNPs), $r_{jk}^2$ is the squared correlation between SNPs $j$ and $k$ in the HapMap3 reference panel and $1 + \text{Na}$ is the LD score regression intercept for trait $j$.

To determine whether a particular realization is significantly larger than expected (and thus the ratio $\chi^2_{\text{observed}}/\chi^2_{\text{expected}}$ is significantly greater than one), we test each particular observed $Z$ statistic (the square root of the $\chi^2$) for SNP $j$ against a normal distribution with variance: $(N_j \times h_j^2 \times \text{LDscore}_i/M) + (1 + \text{Na})_j$. We used precomputed LD scores available from the LDSC software[38]. As recommended by Bulik-Sullivan et al.[38], we restricted our analysis to HapMap3 SNPs (using the merge-alleles flag) because these seem to be well-imputed in most studies. Out of 132 SNPs with $P_{\text{EA}} < 1 \times 10^{-5}$ and $P_{\text{SZ}} < 0.05$, only 30 SNPs are directly present in HapMap3 SNP list (Supplementary Data 9). Therefore, we extracted proxy SNPs with $r^2 > 0.8$ and a maximum distance of 500 kb to our missing EA lead SNPs and chose the one with the highest $r^2$ as a proxy. After this step, we could include 105 (out of 132) SNPs in our analyses. For each of these 105 SNPs, we observed the $Z$-statistics in the publicly available GWAS results of the traits. $Z$-statistics were converted into $\chi^2$ statistics by squaring them. The LD score corrected enrichment per SNP for each trait is the ratio of the observed to the expected $\chi^2$.

Furthermore, since the SNPs considered for enrichment are independent, we can use Fisher's method to combine the enrichment $P$ values per SNP into a single $P$ value per trait. The latter $P$ value reflects excess enrichment for the set of SNPs beyond what is expected if these SNPs are part of the infinitesimal genetic contribution to the trait in question.

The results are shown in Supplementary Fig. 5 (and in Supplementary Data 10).

Our LD-aware enrichment test has two limitations. First, LD score regression assumes that allele frequency (AF) does not correlate with effect size, an assumption which has been empirically shown to be violated for low-frequency alleles[56]. Second, our test assumes the absence of selection on the trait. Variation in AF and the degree of negative selection could explain excess signal in low LD SNPs[57]. However, our raw enrichment $P$ value is robust to this because it takes the AF of the candidate SNPs explicitly into account.

**Replication of PPM results in the GRAS data collection**. The GRAS data collection complies with the Helsinki Declaration and were approved by the Ethics Committees of the Universities of Göttingen and Greifswald, Germany or of collaborating centres. All subjects and/or their authorized legal representatives gave written informed consent.

We showed in our pre-registered analysis plan that our replication sample (GRAS) is not large enough to replicate individual SNPs (https://osf.io/dnhfk/). Instead, we decided at the outset to attempt replication of the proxy-phenotype analysis results using a PGS that consists of the >80 most strongly associated, independent SNPs. The set that best meets this criterion are the 132 independent EA lead SNPs that are also nominally associated with SZ ($P_{\text{SZ}} < 0.05$). PGS for this set of 132 candidate SNPs were constructed using either the $\beta$ coefficient estimates of the EA or the SZ GWAS meta-analysis, resulting in two different scores (named *EA_132* and *SZ_132*).

In addition, we also constructed PGS for EA, SZ, BIP and neuroticism in the GRAS data collection using all available SNPs as control variables for multivariate prediction analyses (named *EA_all*, *SZ_all*, *BIP_all* and *Neuro_all*). Technical details are described below.

**Polygenic score calculations in the GRAS data collection**. PGS were calculated using PLINK version 1.9[44,45]. We calculated eight different scores, which are described below. Supplementary Fig. 6 shows the distribution of the MAFs of all SNPs that were included in these PGS.

SZ scores: We received the GWAS summary statistics for SZ from the PGC excluding the data from our replication sample (GRAS). We constructed a PGS using the 132 EA lead SNPs ($P_{\text{EA}} < 10^{-5}$) that are also nominally associated with SZ ($P_{\text{SZ}} < 0.05$). This score (*SZ_132*) is used for replication of the proxy-phenotype analyses.

Furthermore, we constructed a PGS using all 8,240,280 SNPs that survived quality control (*SZ_all*). Next, we applied the clumping procedure using $r^2 > 0.1$ and 1000 kb as the clumping parameters and the 1000 Genomes phase 1 version 3 European reference panel to estimate LD among SNPs, eventually leaving a set of 349,357 SNPs ready for profile scoring. The vast majority of the SNPs we included in the SZ PGS (79%) have MAF > 1% and MAF < 99% (Supplementary Fig. 6).

For secondary analyses on the prediction of SZ symptoms, we constructed two PGS using the 349,357 SNPs that have concordant (+ and +; or – and –) or discordant signs (+ and –; or – and +) for EA and SZ, respectively. This resulted in 174,734 and 174,623 independent SNPs with concordant or discordant, respectively. We used these approximately independent SNPs for profile scoring and call the resulting PGS *Concordant* and *Discordant*. We note that this approach of constructing the *Concordant* and *Discordant* scores is based on a very conservative LD-pruning algorithm because we first LD-pruned all SNPs that survived quality control for both phenotypes and then sort them based on sign concordance. Thus, this approach prunes for LD both within and across the *Concordant* and *Discordant* scores.

As an alternative, we also used a less conservative approach that only prunes for LD within scores. Specifically, this alternative approach first sorts all SNPs that pass quality control for both phenotypes (8,240,280 SNPs) based on sign concordance (4,147,926 SNPs with concordant and 4,092,354 SNPs with discordant signs) and then LD-prunes the resulting set, yielding 260,441 and 261,062 independent SNPs with concordant or discordant, respectively. We call the PGS resulting from this approach *Concordant_more_SNPs* and *Discordant_more_SNPs*.

To enable tests for the sensitivity of our results to the MAF distribution of SNPs included in our scores, we also constructed PGS using SNPs with MAF > 1% (i.e. dropping SNPs with allele frequency ≤0.01 or ≥0.99) and with MAF > 10% (i.e. dropping SNPs with allele frequency ≤0.10 or ≥0.90), respectively, from the 8,240,280 SNPs that survived quality control. Next, we used $r^2 > 0.1$ and 1000 kb as the clumping parameters and the 1000 Genomes phase 1 version 3 European reference panel to estimate LD among SNPs, eventually leaving a set of 302,150 approximately independent lead-SNPs with MAF > 1% and 108,075 SNPs with MAF > 10% ready for profile scoring. The resulting scores are called *SZ_all_1%* and *SZ_all_10%*, respectively. We also constructed PGS from this set of SNPs that took the sign concordance between EA and SZ into account and call these PGS *Concordant_1%* (151,265 SNPs), *Discordant_1%* (150,885 SNPs) and *Concordant_10%* (54,287 SNPs), *Discordant_10%* (53,788 SNPs), respectively.

All scores were calculated using the *–score function* in PLINK using the natural log of the OR of the SNPs for SZ as effect sizes.

Educational attainment scores: Beta coefficients for the EA GWAS were approximated using $\widehat{\beta}_j = \frac{z_j}{\sqrt{N_j * 2 * MAF_j * (1 - MAF_j)}}$, see Rietveld et al.[47] for the derivation. Using these betas, we constructed a PGS using the 132 EA lead SNPs ($P_{EA} < 1 \times 10^{-5}$) that are also nominally associated with SZ ($P_{SZ} < 0.05$). The resulting score is called *EA_132*.

Furthermore, we constructed a PGS using all 8,240,280 SNPs that survived quality control. Next, we applied the clumping procedure using $r^2 > 0.1$ and 1000 kb as the clumping parameters and the 1000 Genomes phase 1 version 3 European reference panel to estimate LD among SNPs, eventually leaving a set of 348,429 SNPs ready for profile scoring (Supplementary Fig. 6). The resulting score is called *EA_all*. Eighty percent of the SNPs we included in the EA PGS have MAF > 1% & MAF < 99% (Supplementary Fig. 6).

We also constructed PGS using SNPs with MAF > 1% i.e. dropping SNPs with allele frequency ≤0.01 or ≥0.99) and with MAF > 10% (i.e. dropping SNPs with allele frequency ≤0.10 or ≥0.90), respectively, from the 8,240,280 SNPs that survived quality control. Next, we used $r^2 > 0.1$ and 1000 kb as the clumping parameters and the 1000 Genomes phase 1 version 3 European reference panel to estimate LD among SNPs, eventually leaving a set of 306,977 approximately independent lead-SNPs with MAF > 1% and 106,607 SNPs with MAF > 10% ready for profile scoring. The resulting scores are called *EA_all_1%* and *EA_all_10%*, respectively.

BIP score: We obtained GWAS summary statistics on BIP from the PGC[58]. We used the LD-pruned GWAS summary from PGC ('pgc.bip.clump.2012–04.txt') with a set of 108,834 LD-pruned SNPs ready for profile scoring. PGS for the GRAS data collection were calculated by the application of the *–score function* in PLINK using the natural log of the OR. The resulting score is called *BIP_all*.

Neuroticism score: We obtained GWAS summary statistics on Neuroticism from the SSGAC. The results are based on the analyses reported in Okbay et al.[26] containing 6,524,432 variants. We applied the clumping procedure using $r^2 > 0.1$ and 1000 kb as the clumping parameters and the 1000 Genomes phase 1 version 3 European reference panel to estimate LD among SNPs, eventually leaving a set of 232,483 SNPs ready for profile scoring (Supplementary Fig. 6). PGS for the GRAS data collection were calculated by the application of *–score function* in PLINK using the Neuroticism beta values. The resulting score is called *Neuro_all*.

Note that our replication sample (GRAS) was not included in the GWAS summary statistics of any of these traits.

Polygenic score correlations: We calculated Pearson correlations between all PGS that we constructed in the GRAS data collection (*SZ_all, SZ_132, EA_all, EA_132, Concordant, Discordant, BIP_all* and *Neuro_all*). Results for SZ patients and healthy controls together are reported in Supplementary Data 12a. We found very similar results among the SZ cases (Supplementary Data 12b) and healthy controls when we analysed them separately from each other (Supplementary Data 12c). These results were used to inform the correct multiple regression model specification for the polygenic prediction analyses (Supplementary Note 6 and 8).

**GWIS**. A GWIS infers genome-wide summary statistics for a (non-linear) function of phenotypes for which GWAS summary statistics are available[41]. Here, in particular, we wish to infer for each SNP the effect on SZ, conditioned upon its effect on BIP. One possible approximation involves a GWIS of the following linear regression function:

$$SZ = \beta * BIP + e$$

where the parameter $\beta$ is estimated from the genetic covariance between SZ and BIP and the genetic variance in BIP as $\beta = \frac{cov_g(SZ, BIP)}{var_g(BIP)}$. The residual ($e$) is actually our trait of intrest, for which we use the term $SZ_{(min\ BIP)}$. Using GWIS we infer the genome-wide summary statistics for $SZ_{(min\ BIP)}$ given the most recent PGC GWAS results for SZ (omitting the GRAS data collection)[5] and BIP[59]. The effect size with respect to $SZ_{(min\ BIP)}$ for a single SNP is computed as:

$$eff_{sz} - \beta * eff_{BIP} = eff_e$$

The standard error for each SNP effect is approximated using the delta method and accounts for the possible effect of sample overlap between the SZ and BIP GWAS.

As data input, we used the GWAS results on SZ (excluding the GRAS data collection). GWAS results for BIP[59] (6990 cases; 4820 controls) were obtained from the website of the PGC (https://www.med.unc.edu/pgc/files/resultfiles/pgc.cross.bip.zip).

Using the same method and data, we also 'purged' the genetic association results for BIP of their overlap with SZ, obtaining 'unique' $BIP_{(min\ SZ)}$ results.

**Code availability**. Source code for GWIS and LD-aware enrichment analyses are available at https://github.com/MichelNivard/EA_SZ.

**Data availability**. The GWAS summary statistics that support the findings of this study are available on the website of the SSGAC: http://www.thessgac.org/#!data/

kuzq8. The GRAS data collection is not publicly available due to data protection laws in Germany that strictly safeguard the privacy ofstudy participants. To request access, contact the study's principal investigator Prof. Dr. Hannelore Ehrenreich (ehrenreich@em.mpg.de)

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

## Acknowledgements

This research was carried out under the auspices of the Social Science Genetic Association Consortium (SSGAC), including use of the UK Biobank Resource (application reference number 11425). We thank all research consortia that provide access to GWAS summary statistics in the public domain. Specifically, we acknowledge data access from the Psychiatric Genomics Consortium (PGC), the Genetic Investigation of ANthropometric Traits Consortium (GIANT), the International Inflammatory Bowel Disease Genetics Consortium (IIBDGC), the International Genomics of Alzheimer's Project (IGAP), the CARDIoGRAMplusC4D Consortium, the Reproductive Genetics Consortium (ReproGen), the Tobacco and Genetics Consortium (TAG), the Meta-Analyses of Glucose and Insulin-related traits Consortium (MAGIC), the ENIGMA Consortium and the Childhood Intelligence Consortium (CHIC). We would like to thank the research participants and employees of 23andMe for making this work possible as well as Joyce Y. Tung, Nicholas A. Furlotte and David A. Hinds from the 23andMe research team. This study was supported by funding from an ERC Consolidator Grant (647648 EdGe, Philipp D. Koellinger), the Max Planck Society, the Max Planck Förderstiftung, the DFG (CNMPB), EXTRABRAIN EU-FP7, the Niedersachsen-Research Network on Neuroinfectiology (N-RENNT) and EU-AIMS. Michel G. Nivard was supported by a Royal Netherlands Academy of Science Professor Award to Dorret I. Boomsma (PAH/6635). Additional acknowledgements are provided in the Supplementary Note 12.

## Author contributions

P.D.K. designed and oversaw the study and conducted proxy-phenotype analyses. V.B. and M.M. carried out analyses in the GRAS sample. R.d.V., C.A.P.B., M.G.N. and P.D.K. developed statistical methods. V.B. conducted bioinformatics and computed the LD-aware enrichment tests. C.A.P.B. and R.d.V. conducted simulation analyses. M.G.N. computed GWIS results, genetic correlations and carried out pleiotropy analyses. R.K.L. assisted with biological annotation and visualization of results. P.D.K., V.B., M.M. and H. E. made especially major contributions to writing and editing. All authors contributed to and critically reviewed the manuscript.

## Additional information

**Competing interests:** The authors declare no competing interests.

