## [Peer Review File · Nature Communications]

Reviewer #1 (Remarks to the Author):

This is important work on the leading edge of interpretation of large scale gwas, and I would be eager to publish the paper in ncomms. The utility of genetic scores that are composite of multiple phenotype gwas (whether scz,ea PRS or gwis genetic correlations), are of great interest to a general audience. The methods developed and analyses performed are a tour de force. The finding of genetic scores that correlate with scz severity or endophenotypes is important to the field, but needs more characterization and [to be interpreted as needing] replication. Most importantly, the paper needs a lot of writing work still.

Specifically, the methods and supplement are beautifully written and crafted; however, the results and discussion are not to say the least. In a journal like ncomms (in my view), the results should provide some concise description of methods that makes it possible to understand without constant reference to the very in depth methods and supp. (For comms, will the methods appear after results? If so, then the paper is not correctly formatted.) This issue gets progressively worse to the point that the last paragraphs of results are single sentences with little context, justification or connection to the rest of the paper. This continues in the discussion which is extremely short and underdeveloped. The last sentence of Discussion seems to come out of nowhere.

I am critical of the premise of enriched but not correlated gwas reflecting genetic heterogeneity, but I accept it as the authors' novel and interesting premise. Of course the idea of differentiating between genetic dependence and correlation is well stated and clearly is the case for EA and SZ. This could be a semantic issue, by genetic heterogeneity I mean the idea that there are subtypes of scz that might be treated differently, and one might suggest to the authors to carefully use the phrase "genetic heterogeneity among scz cases" rather than "genetic heterogeneity of SZ" or even "in SZ" (which is ambiguous). It increases the readers tendency to be critical, that clear genetic heterogeneity of EA (part IQ and part not) is not dealt with in this paper, and that one method (sign con/discordant PRS) is used in some analyses and another (GWIS) is used in others. Still, I applaud the authors for the exposition of the problem and for Supplementary section 1 in particular; this is a great contribution to the complex trait genetics literature.

There are some analyses/results that need better explanation and justification. These include the Bayesian analysis of scz association given scz and ea p-vals, the categorization as pleiotropic of loci with shared ea and scz causal variant posterior probs, and, especially given the manuscripts emphasis on it, the prediction of scz subphenotypes being not overwhelmingly significant and possibly driven or at least influenced by phenotypic correlations within their test data. Without suggesting substantial more work, the authors should discuss more the limitations to the results and the need for further validation/replication. These are numerous analyses, and the paper could drop some of them without substantial loss of impact.

Specific issues:

Especially given the manuscripts emphasis on it, the prediction of scz subphenotypes being not overwhelmingly significant and possibly driven or at least influenced by phenotypic correlations within their test data may limit the impact of these results, and should be noted in discussion. Are the same phenotypic correlations seen in other datasets/supported by the literature? Are the correlations of sign con/discordant PRS with GAF and Negative symptoms robust to including eduYr or Premorbid IQ in the model?

The categorization as pleiotropic of loci with shared ea and scz causal variant posterior probs does not seem necessarily well-supported. I think the results table is missing, I can't find it (perhaps it should be added to Table 1 [perhaps with fewer Bayesian probs?]). The overlapping associations could still be due to linkage if these loci have high LD and large credible sets; therefore, these statistics for the loci should also be reported and considered in the interpretation of results. Also the enrichment of associations controlling for LD-score does not in itself rule out linkage, so does not support pleiotropy over linkage, as stated.

The gwis analysis of scz and bipolar, while of great interest, seems disconnected from the PGS analyses and not described well in Results. Why not use gwis-based PGS instead of sign concordance for EA,SZ? Why not make some BIP,SZ gwis-based polygenic scores and test them on subphenotypes?

Reviewer #2 (Remarks to the Author):

Bansal et al aim to assess the role of genetic heterogeneity in schizophrenia. Toward this aim they first use the proxy phenotype method developed by Rietveld et al. to identify 132 educational attainment (EA) SNPs that are associated with schizophrenia (SZ) at the nominal significance level. Among 21 SNPs which survived Bonferroni correction, 12 were in LD with SNPs previously reported to be associated with SZ and 8 are predicted to have direct causal effects on both EA and SZ (PAINTOR tool).

To biologically annotate loci shared between EA and SZ, Bansal et al applied the DEPICT tool on the 132 SNPs shared SNPs. They identified 19 approximately independent brain-related reconstituted gene sets related, prioritized tissues related to the CNS and prioritized several genes some of which have been implicated in cognitive processes related to memory and emotion.

Finally, using the “Göttingen Research Association for Schizophrenia” (GRAS) cohort, Bansal et al show that a polygenic risk score comprising SZ SNPs, which are sign concordant with respect to EA, increase predictive accuracy for the severity of SZ. Specifically, they show that individuals with higher genetic propensity for EA have less severe negative SC symptoms.

Genetic association results provide a robust starting point to unravel shared genetics and biology between traits and disorders. Bansal et al.’s work is original and especially the part based on the polygenic risk scores applied on the GRAS cohort provides findings potentially very important to understanding the complicated relationship between educational attainment and schizophrenia. I acknowledge that this may be beyond the scope of this paper, but going forward it will become extremely interesting to identify the specific brain areas and cell types underlying the different SZ subtypes. Using e.g. DEPICT with single cell RNA-seq data from relevant brain areas could potentially provide some first answers to this question.

Reviewer #3 (Remarks to the Author):

This is an interesting paper, aiming to explore a counter-intuitive finding: correlation between EA and SZ. The study is well designed, and the results are interesting but the story is incomplete. Three suggestions:

1. Look at the overlap with bipolar SNPs also, as the SNPs that are overlapping in the same direction in SZ and EA may be bipolar overlap SNPs.
2. Focus on positive symptoms SZ with these SNPs.
3. Do pathway analyses with them, see if pathways related to energy metabolism and neurotrophic factors are involved.

Response to reviewer remarks

We thank the reviewers for their constructive comments. Below, we respond to each of them. The original remarks are marked in bold.

Reviewer #1 (Remarks to the Author):

This is important work on the leading edge of interpretation of large scale gwas, and I would be eager to publish the paper in ncomms. The utility of genetic scores that are composite of multiple phenotype gwas (whether scz,ea PRS or gwis genetic correlations), are of great interest to a general audience. The methods developed and analyses performed are a tour de force. The finding of genetic scores that correlate with scz severity or endophenotypes is important to the field, but needs more characterization and [to be interpreted as needing] replication. Most importantly, the paper needs a lot of writing work still.

Specifically, the methods and supplement are beautifully written and crafted; however, the results and discussion are not to say the least. In a journal like ncomms (in my view), the results should provide some concise description of methods that makes it possible to understand without constant reference to the very in depth methods and supp. (For comms, will the methods appear after results? If so, then the paper is not correctly formatted.) This issue gets progressively worse to the point that the last paragraphs of results are single sentences with little context, justification or connection to the rest of the paper. This continues in the discussion which is extremely short and underdeveloped. The last sentence of Discussion seems to come out of nowhere.

We thank the referee for the positive remarks and the suggestions. The reason for the unusual format of our original submission and the overly short discussion section was that our paper was forwarded to Nature Communications from another journal that had stricter space limits and different formatting requirements. We have reformatted our paper according to Nature Communication's guidelines, integrated a brief description of the main methods into the Results section, and expanded the discussion of our findings. We also mention specifically that further replication of our results would be highly desirable.

I am critical of the premise of enriched but not correlated gwas reflecting genetic heterogeneity, but I accept it as the authors' novel and interesting premise. Of course the idea of differentiating between genetic dependence and correlation is well stated and clearly is the case for EA and SZ. This could be a semantic issue, by genetic heterogeneity I mean the idea that there are subtypes of scz that might be treated differently, and one might suggest to the authors to carefully use the phrase "genetic heterogeneity among scz cases" rather than "genetic heterogeneity of SZ" or even "in SZ" (which is ambiguous).

We thank the referee for this suggestion and we agree that the term "genetic heterogeneity" may imply different biological subtypes of a disease that may need to be treated differently. However, it also conceivable that different biological subtypes of a disease may still react to some treatments in a similar way (e.g. a better environment for patients). Obviously, the results we present here are only a modest step towards unravelling the complexity of schizophrenia (SZ) and much more research would

need to be done to draw conclusions about the biological processes that are involved and their potentially varying response to treatment.

Regarding the terminology we use in the paper, we considered the suggestion to talk about “genetic heterogeneity among SZ cases”, but thought that this may also be misunderstood: Ultimately, any two patients will be genetically different from each other unless they are monozygotic twins, but this type of “genetic heterogeneity among SZ cases” is not informative about the nature of the disease. In the end, we decided to keep our original terminology, but we tried to clarify its meaning and implications throughout the revised manuscript.

It increases the readers tendency to be critical, that clear genetic heterogeneity of EA (part IQ and part not) is not dealt with in this paper, and that one method (sign con/discordant PRS) is used in some analyses and another (GWIS) is used in others.

We apologize for the confusion. The reason why we did not discuss the genetic heterogeneity of EA in more detail in the original submission is that these analyses were reported in a previous paper (Okbay et al. 2016) and that space constraints forced us to abstain from a more detailed description and discussion. We have remedied this in the revised version and now discuss these analyses and their implications in the main text.

We have also clarified the different, complementary purposes of using both GWIS and the PRS analyses in the revised paper. Specifically, the concordant / discordant PGS are used to test for genetic heterogeneity of SZ.

The GWIS analyses serve a different purpose: The aim of that analysis is to delineate the genetic architecture of SZ and bipolar disorder (BIP) from each other to explore the hypothesis that (unique) SZ is a neurodevelopmental disorder while (unique) BIP is not¹⁻⁵. We can triangulate the correct answer by looking at the genetic components that are unique to SZ and BIP and by testing the genetic correlations of these unique components with educational attainment (EA) and childhood intelligence quotients. The results of this analysis support the idea that the differentiating factor between SZ and BIP is indeed that the former is partly a neurodevelopment disorder while the latter is not. This adds to the overall impression that current SZ diagnoses aggregate over possibly quite different disease aetiologies.

Still, I applaud the authors for the exposition of the problem and for Supplementary section 1 in particular; this is a great contribution to the complex trait genetics literature.

Thank you.

There are some analyses/results that need better explanation and justification. These include the Bayesian analysis of scz association given scz and ea p-vals, the categorization as pleiotropic of loci with shared ea and scz causal variant posterior probs, and, especially given the manuscripts emphasis on it, the prediction of scz subphenotypes being not overwhelmingly significant and possibly driven or at least influenced by phenotypic correlations within their test data. Without suggesting substantial more work, the authors should discuss more the limitations to the results and the need for further validation/replication. These are numerous analyses, and the paper could drop some of them without substantial loss of impact.

In the revised version of the paper, we added a more detailed discussion of the Bayesian probabilities of specific loci being truly associated with SZ. We also extended our discussion of the analyses that gauge the possibility of direct pleiotropic effects of a locus on both traits.

The prediction of the SZ subphenotypes is indeed not very strong in our main analysis, but this is primarily a result of the extremely conservative LD-pruning algorithm our test uses. In particular, to construct the set of concordant and discordant SNPs, we first LD-prune all SNPs that survived QC for

both phenotypes and then sort them based on sign concordance. This approach results in only 174,734 and 174,623 independent SNPs with concordant or discordant, respectively.

We now also conduct an alternative approach that maintains more signal in each score. Specifically, we first sort all SNPs that pass QC for both phenotypes based on sign concordance and then LD-prune the resulting set. This approach yields 260,441 and 261,062 independent SNPs with concordant or discordant, respectively (**Supplementary Note 11.1.1**). Using these scores in the prediction of SZ measures yields much better results (**Supplementary Table 8.7**). In particular, the polygenic prediction of GAF measured in ΔR^2 improves from 0.39% in the main model to 0.82% in the alternative specification. Similarly, predictive accuracy for PANSS positive and negative improves from 0.17% ($p_F = 0.098$) and 0.63% ($p_F = 0.007$) to 0.45% ($p_F = 0.016$) and 1.12% ($p_F = 0.0004$), respectively.

Despite the better prediction results with the alternative approach, we still prefer our main approach because it is more conservative and a direct implementation of the statistical heterogeneity test we developed in **Supplementary Note 1**. In the revised paper, we added the alternative approach to **Supplementary Notes 11.1.1** and discuss the additional results both in **Supplementary Note 12.3** and the main text.

We address the issue of phenotypic correlation in our reply to the following comment.

We considered the suggestion to drop some of the analyses from the paper. While we agree that some analyses are certainly more important than others, we decided in the end to keep all analyses in the paper because we believe that all of them help to interpret the patterns we see in the data. The discussion of the less central analyses is largely relegated to the SI.

We now also mention in the discussion section of the main text that further replication of our results would be highly desirable.

Specific issues:

Especially given the manuscripts emphasis on it, the prediction of scz subphenotypes being not overwhelmingly significant and possibly driven or at least influenced by phenotypic correlations within their test data may limit the impact of these results, and should be noted in discussion. Are the same phenotypic correlations seen in other datasets/supported by the literature? Are the correlations of sign con/discordant PRS with GAF and Negative symptoms robust to including eduYr or Premorbid IQ in the model?

We discussed the reasons for the limited predictive accuracy of the SZ subphenotypes above, and also added this discussion and our new, stronger results to the paper and the SI.

To investigate if the phenotypic correlations in the GRAS sample are comparable to patterns in other datasets, we compare them to findings from a recent meta-analysis by Suzuki et al. 2015⁶ who examined the correlations between different clinical outcomes in SZ patients, specifically the correlations between GAF, CGI-S and PANSS. Table R1 below compares the results reported by Suzuki and colleagues with those we find in the GRAS sample. Overall, the correlation patterns in the GRAS sample resemble those reported by Suzuki et al., except that the correlations between GAF and the PANSS scales is slightly stronger in the GRAS sample. We now mention this in **Supplementary Note 12.1**, page 29.

Table R1 – Comparison of phenotypic correlations between GRAS and other samples

Correlated phenotypes	Suzuki et al. 2015	GRAS sample
GAF - CGI	38 studies; n=11315 Pearson's $r=-0.893$; $p<0.0001$	Pearson's $r=-0.824$; $p<0.0001$
GAF - PANSS	GAF - PANSS Total 40 studies; n=8000 Pearson's $r=-0.401$; $p<0.0001$	GAF - PANSS Positive Pearson's $r=-0.629$; $p<0.0001$ GAF - PANSS Negative Pearson's $r=-0.579$; $p<0.0001$
CGI - PANSS	CGI - PANSS Total 25 studies; n=5438 Pearson's $r=-0.692$; $p<0.0001$	CGI - PANSS Positive Pearson's $r=-0.638$; $p<0.0001$ CGI - PANSS Negative Pearson's $r=-0.598$; $p<0.0001$

We followed the reviewer's advice and conducted additional predictions of SZ subphenotypes (GAF, CGI-S, PANSS positive, PANSS negative) controlling for years of education or premorbid IQ, respectively. We report these robustness checks in **Supplementary Note 12.3** and in **Supplementary Tables 8.6.a** and **8.6.b**. The bottom line is that the increase in predictive accuracy from splitting the score by sign concordance is robust to controlling for years of education and premorbid IQ, even though the inclusion of these covariates leads to a reduction of sample size due to missing data, especially in the case of premorbid IQ.

The categorization as pleiotropic of loci with shared ea and scz causal variant posterior probs does not seem necessarily well-supported. I think the results table is missing, I can't find it (perhaps it should be added to Table 1 [perhaps with fewer Bayesian probs?]). The overlapping associations could still be due to linkage if these loci have high LD and large credible sets; therefore, these statistics for the loci should also be reported and considered in the interpretation of results.

We reported the results of this analysis in **Supplementary Table 3** and now also added them to **Table 1** in the main text. The revised version of the paper discusses our fine-mapping approach, its shortcoming and results in greater detail. Several loci have large credibility sets, which indicates the limits of statistical fine mapping in the current sample. However, one locus (rs7336518) has medium credibility and a small credibility set (18 SNPs for EA and 2 SNPs for SZ), which makes it a reasonable candidate for having direct pleiotropic effects on both traits. Further progress could be made with trans-ethnic fine-mapping (and would require trans-ethnic GWAS of EA and SZ), sequence based fine-mapping in close relatives, or experimental fine-mapping, all of which are beyond the scope of our current study. We maintain, however, that the statistical fine-mapping results we reported increase available information. Furthermore, at least for loci with medium or high ratings, their posterior probability of being causal for both traits seem more likely than before our analysis.

Also the enrichment of associations controlling for LD-score does not in itself rule out linkage, so does not support pleiotropy over linkage, as stated.

We clarified in the revised version what this test can and cannot achieve. In particular, the LD-aware enrichment test we developed is meant to distinguish a "true" conditional marginal signal (significant for SZ given a threshold for EA) from a "spurious" marginal conditional signal that is entirely induced by very high LD. Of course, ideally, we would want to distinguish conclusively between pleiotropy and LD, but this test cannot do that. Yet, we believe it is still an improvement compared to the "raw" enrichment tests that ignore LD entirely.

The gwis analysis of scz and bipolar, while of great interest, seems disconnected from the PGS analyses and not described well in Results. Why not use gwis-based PGS instead of sign concordance for EA,SZ? Why not make some BIP,SZ gwis-based polygenic scores and test them on subphenotypes?

We believe that the GWIS analyses add value to the paper by demonstrating that a part of SZ seems to be a neurodevelopmental disorder (i.e. the part that is genetically different from BD), while other parts are not (i.e. the parts that are genetically overlapping with BD). This adds to the overall impression that SZ may be a diagnoses that spans over multiple disease etiologies. In the revised version of the paper, we discuss this in greater detail.

The reason we did not construct GWIS-based polygenic scores is statistical power: When GWIS is used to “deconvolute” two traits, the uncertainty in the GWIS estimate for individual SNPS is a function of the uncertainty of the SNP effect as observed in two studies. Because the GWAS on BIP was conducted in a relative small sample size, a PRS constructed from the GWIS results would suffer from betas that are estimated with great uncertainty. As a result, the expected score would not be expected to predict well (or predict at all). As the BIP sample size will increase in the future, GWIS and GWIS-derived polygenic scores will increase in utility.

Reviewer #2 (Remarks to the Author):

Bansal et al aim to assess the role of genetic heterogeneity in schizophrenia. Toward this aim they first use the proxy phenotype method developed by Rietveld et al. to identify 132 educational attainment (EA) SNPs that are associated with schizophrenia (SZ) at the nominal significance level. Among 21 SNPs which survived Bonferroni correction, 12 were in LD with SNPs previously reported to be associated with SZ and 8 are predicted to have direct causal effects on both EA and SZ (PAINTOR tool).

To biologically annotate loci shared between EA and SZ, Bansal et al applied the DEPICT tool on the 132 SNPs shared SNPs. They identified 19 approximately independent brain-related reconstituted gene sets related, prioritized tissues related to the CNS and prioritized several genes some of which have been implicated in cognitive processes related to memory and emotion.

Finally, using the “Göttingen Research Association for Schizophrenia” (GRAS) cohort, Bansal et al show that a polygenic risk score comprising SZ SNPs, which are sign concordant with respect to EA, increase predictive accuracy for the severity of SZ. Specifically, they show that individuals with higher genetic propensity for EA have less severe negative SC symptoms.

Genetic association results provide a robust starting point to unravel shared genetics and biology between traits and disorders. Bansal et al.’s work is original and especially the part based on the polygenic risk scores applied on the GRAS cohort provides findings potentially very important to understanding the complicated relationship between educational attainment and schizophrenia. I acknowledge that this may be beyond the scope of this paper, but going forward it will become extremely interesting to identify the specific brain areas and cell types underlying the different SZ subtypes. Using e.g. DEPICT with single cell RNA-seq data from relevant brain areas could potentially provide some first answers to this question.

We thank the referee for the positive remarks. We agree that a more fine-grained bio-annotation of the different SZ subtypes would be very interesting. We considered possibilities of making progress along these lines with the currently available data, but came to the conclusion that it is better to wait with such analyses until additional data becomes available. For example, one way forward would be a more detailed bio-annotation which contrasts discordant and concordant SNPs that are strongly associated with both EA and SZ. However, this currently yields only small subsets of SNPs that limit the power of bio-annotation tools such as DEPICT severely. The next generation of GWAS results on EA and SZ, which will be based on substantially larger sample sizes that are expected to yield a much higher number of jointly associated SNPs, should enable such analyses.

Reviewer #3 (Remarks to the Author):

This is an interesting paper, aiming to explore a counter-intuitive finding: correlation between EA and SZ. The study is well designed, and the results are interesting but the story is incomplete.

We thank the referee for the positive remarks and the suggestions below.

Three suggestions:

- 1. Look at the overlap with bipolar SNPs also, as the SNPs that are overlapping in the same direction in SZ and EA may be bipolar overlap SNPs.**

In the original submission, we started to investigate the genetic overlap between EA, SZ, and BIP in the context of the LD-aware enrichment tests (**Supplementary Note 9, Supplementary Figure 4, Supplementary Table 5.1**). Indeed, there is clear evidence for joint enrichment across all three traits. Furthermore, the results of our GWIS analyses (**Supplementary Note 13, Figure 3**) are clearly in line with the referee's suggestion that concordant SNPs for EA and SZ tend to be positively associated with BIP.

However, there are several reasons why the original submission did not investigate this link further along the lines suggested by the referee:

- The original BIP GWAS was relatively small and had limited statistical power (6990 cases, 4820 controls).
- There was sample overlap between BIP and SZ, as well as between BIP and EA. This makes it difficult to interpret the results of such a three-way look-up⁷⁻⁹.

Nevertheless, we now conducted these analyses, as suggested by the referee.

As a first step, we restricted our GWAS summary statistics to HapMap3 SNPs because these tend to be well-imputed in most studies. Out of 132 SNPs with $P_{EA} < 1 \times 10^{-5}$ and $P_{SZ} < 0.05$, only 30 SNPs are directly present in HapMap3 SNP list. Therefore, we extracted proxy SNPs with $r^2 > 0.8$ and a maximum distance of 500 kb to our missing EA lead SNPs and chose the one with the highest r^2 as a proxy. After this step, we could include 105 (out of 132) SNPs in our analyses. Out of these, 93 are present in the BIP summary statistics from the PGC. There are 29 SNPs that are concordant for EA, SZ and BP, see Table R2 below.

Table R2: Sign-concordant SNPs for SZ and EA ($P_{EA} < 1 \times 10^{-5}$ and $P_{SZ} < 0.05$) or their proxies that also have sign-concordant effects with BIP

SNP	SZ Odds	SZ P-value	BIP Odds	BIP P-value
rs10219183	0.98030001	0.1177	0.8951	8.992e-05
rs10486428	0.92219001	2.341e-06	0.9009	0.003568
rs10514857	0.96977001	0.0048389998	0.9636	0.1105
rs10748815	0.96310002	0.00056189997	0.9742	0.2712
rs11078827	0.97531003	0.020470001	0.9944	0.8113
rs11100197	0.97316998	0.01356	0.9592	0.0767
rs11191527	0.95858002	0.003764	0.9937	0.8505
rs11588857	1.04436	0.0007181	1.0451	0.1181
rs12202969	1.03593	0.0009557	1.0818	0.0006571
rs12234369	1.02521	0.03404	1.0711	0.006645
rs12739630	0.96714997	0.01331	0.9332	0.0187
rs13093396	0.96397001	0.00061809999	0.9963	0.8738
rs13176430	1.0277801	0.01063	1.0079	0.7358
rs1461356	1.04196	0.000149	1.0849	0.0004803
rs1609520	0.91860002	2.526e-12	0.9819	0.4976
rs17229260	0.97579998	0.02826	0.992	0.7389
rs17638867	1.03097	0.031490002	1.0087	0.7791
rs17689471	1.05012	0.0004131	1.0416	0.144
rs191692	1.02798	0.01193	1.0197	0.4155
rs193707	1.03521	0.0097939996	1.0464	0.1117
rs2253536	0.96512002	0.0052379998	0.9885	0.6805
rs2490393	0.93221003	3.365e-08	0.9603	0.1398
rs2556370	0.96724999	0.0020029999	0.9405	0.0119
rs3111222	0.97189999	0.01568	0.9575	0.09159
rs562010	0.95714998	0.00074400002	0.9481	0.05681
rs718439	1.02881	0.0087310001	1.024	0.3101
rs756912	0.94800001	5.9630003e-07	0.9916	0.719
rs7790949	1.02306	0.034019999	1.0178	0.4507
rs996234	1.02624	0.016890001	1.029	0.2267

2. Focus on positive symptoms SZ with these SNPs.

The effect size of a polygenic score constructed from the 29 SNPs in Table R2 would most likely be too small to detect effects on positive symptoms in the GRAS cohort (see our preregistered analysis plan for power calculations with realistic effect size assumptions, <https://osf.io/dnhfk/>). Therefore, we proceeded with a more inclusive approach that considers all SNPs that are sign concordant for all three traits. In particular, there are 2,427,220 SNPs in the publically available BIP GWAS results. We merged all EA and SZ from our analyses with the BIP results, resulting in 2,346,462 SNPs. Next, we applied the clumping procedure using $r^2 > 0.1$ and 1,000 kb as parameters and the 1000 Genomes phase 1 version 3 European reference panel to estimate LD among SNPs, eventually leaving a set of 130,107 LD-pruned SNPs that are available in all three GWASs. Out of these, there are 33,469 SNPs concordant (same direction) in SZ, EA and BP. We constructed a polygenic score with these SNPs using the $-\log$ score function in PLINK and the natural log of the odds ratio of these SNPs for SZ as the effect size. The resulting score was used to predict positive symptoms in the GRAS cohort, see Table R2 below. This EA, SZ and BP concordant score did not predict positive symptoms (p -value = 0.65).

Table R3: Prediction of positive symptoms using a polygenic score that is sign-concordant for EA, SZ, and BIP

PANSS positive		
Baseline Model		
SZ_all	standardized beta	0.0291
	P value	0.424
BIP_all	standardized beta	0.0196
	P value	0.542
EA_all	standardized beta	-0.0003
	P value	0.992
		Adj. R²
		0.0705
Split Model		
Con_SZ_BIP_EA	standardized beta	0.0185
	P value	0.652
SZ_all	standardized beta	0.0196
	P value	0.640
BIP_all	standardized beta	0.0185
	P value	0.566
EA_all	standardized beta	-0.0044
	P value	0.891
		Adj. R²
		0.0698
		n
		1,009
ΔR^2 (Split Model – Baseline Model)		-0.00075

Notes: Linear regression using the first 10 genetic principal components and medication as control variables.

3. Do pathway analyses with them, see if pathways related to energy metabolism and neurotrophic factors are involved.

Using the 29 SNPs from Table R2 as input, DEPICT identified only 1 significant reconstituted gene set at $FDR < 0.05$, i.e. “impaired contextual conditioning behaviour”.

If we use all 108 SNPs that are sign-concordant for EA, SZ, and BIP, and also have $P < 0.05$ for all three traits as input for DEPICT, we find no significant reconstituted gene set at $FDR < 0.05$.

Nevertheless, as we indicated above, these type of analyses may become more informative with the next generation of GWAS results, which will be based on much larger sample sizes. For the current paper, we decided not to include these underpowered analyses.

References

1. Kahn, R. S. & Keefe, R. S. E. Schizophrenia Is a Cognitive Illness. *JAMA Psychiatry* **70**, 1107 (2013).
2. Kraepelin, E. *Psychiatrie: Ein Lehrbuch für Studierende und Ärzte*. (Verlag von Johann Ambrosius Barth, 1893).
3. Trotta, A., Murray, R. & MacCabe, J. Do premorbid and post-onset cognitive functioning differ between schizophrenia and bipolar disorder? A systematic review and meta-analysis. *Psychol. Med.* **45**, 381–394 (2015).
4. Murray, R. M. *et al.* A developmental model for similarities and dissimilarities between schizophrenia and bipolar disorder. *Schizophr. Res.* **71**, 405–416 (2004).
5. Murray, R. M., O'Callaghan, E., Castle, D. J. & Lewis, S. W. A Neurodevelopmental Approach to the Classification of Schizophrenia. *Schizophr. Bull.* **18**, 319–332 (1992).
6. Suzuki, T. *et al.* Relationships between global assessment of functioning and other rating scales in clinical trials for schizophrenia. *Psychiatry Res.* **227**, 265–269 (2015).
7. Consortium, C.-D. G. of the P. G. Identification of risk loci with shared effects on five major psychiatric disorders: a genome-wide analysis. *Lancet* **381**, 1371–1379 (2013).
8. Ripke, S. *et al.* Biological insights from 108 schizophrenia-associated genetic loci. *Nature* **511**, 421–427 (2014).
9. Okbay, A. *et al.* Genome-wide association study identifies 74 loci associated with educational attainment. *Nature* **533**, 539–542 (2016).

Reviewer #1 (Remarks to the Author):

I stand by my first review that this is a good paper and I am sorry for delay. Actually the authors have mostly not responded to my review with manuscript changes. The biggest responses are that the results and discussion are better written, which is very important (I even get the last bit now about translation), and a caveat paragraph is present for the finemapping analysis which could be better with little work in my opinion see below. The subphenotypes analyses are ok, I understand them better, although I don't really get the high snp numbers in these two scoring approaches (see below).

Three big things I would ask the authors to address:

I still think the question of heterogeneity of ea is begging to be addressed here. Especially the intro seems to emphasize it

I think the gwis analysis is great, and I don't want to ask the authors for more work, but two things.. First, I don't buy the sentence line 375 that the gwis/ldsc results provide evidence of sz heterogeneity at any sort of individual level. If some snps were associated with bip,sz,ea and others are associated only with sz, even while all snps are uniformly associated with any/all subsets of sz patients, would you not get these same results? Second, I just put this out there: I haven't done any real validation analyses of gwis yet, and things in figure 3 like gwis-sz is as similar to bip as it is the sz, and of course gwis-sz,gwis-bip $r_g = -1$, make me wonder if it is doing something biologically interpretable or more just mathematically contrived. If there are some validation analyses you've done that make you feel better about the results, then it might be nice to present them

Third big thing, the cautions of the paintor results are fine, but more could be presented (without much more work) that actually might detract less from the impact than these changes. Importantly, the key here is that both concordant and discordant loci need to have at least some evidence of pleiotropy for the conclusions of the paper to hold, not one or the other. Can the authors make any statement to this effect? Possible additions: 1) I would think 80% or even 50% credible sets could be used. What do their sizes and overlaps look like? 2) I find myself wanting to look at locuszoom plots in ea and sz for the shared loci; usually it's obvious if peaks might be overlapping.

Other things:

To me, if a nature communications paper is a nature or nature genetics paper that's simply not bogged down in endless review, then it might be shorter and more on message. Then, the new loci

and pathway/tissues/genes stuff is totally unnecessary, validation in other genetic data and assortative mating could be single sentences here or there, and the polygenic scoring and gwis stuff is what it's all about. Editorially, could maximize the impact of this paper?

I find the ld aware enrichment section unsatisfying. Overall, my impression is that it's results are the same as the enrichment results with matched snps reported above it, so I don't see why it's not further above or why it's there at all. If depression and related traits are not enriched for sz, then the method lacks power, and the statement of phenotypic specificity is less compelling. How is one snp enriched? Overlap of sz and bip is not new, so is there something new about this result?

Line 190- Odds should be or

Lines 270-280- is ea being tested or sz? Are the enrichments for bip, menarche, with sz or with ea?
Unclear

L 309, one per ld block would lead to far fewer than 300000. Ld pruning would leave closer to 100000 in my experience. What are the parameters here? If there are tons of rare snps, perhaps the analysis would improve by excluding them

Page 9. Is the f test really appropriate here? Should aic or bic be used, since the models are not hierarchical/nested? Honestly this is probably not very important.

Reviewer #3 (Remarks to the Author):

Paper is improved, and additional analyses have been carried out. I would suggest that the data supports the possibility that bipolar -related SNPs mediate the relationship observed between EA and SZ, possibly through higher energy/drive. This possible mediating relationship of Bipolar SNPs makes things less counter-intuitive, and needs to be mentioned more strongly in the discussion and/or abstract.

Response to reviewer remarks

We thank the reviewers for their constructive comments. Below, we respond to each of them. The original remarks are marked in bold.

Reviewer #1 (Remarks to the Author):

I stand by my first review that this is a good paper and I am sorry for delay. Actually the authors have mostly not responded to my review with manuscript changes. The biggest responses are that the results and discussion are better written, which is very important (I even get the last bit now about translation), and a caveat paragraph is present for the finemapping analysis which could be better with little work in my opinion see below. The subphenotypes analyses are ok, I understand them better, although I don't really get the high snp numbers in these two scoring approaches (see below).

We thank the reviewer for the positive feedback. In response to the reviewer's comments on the previous version of the paper, we added further polygenic prediction analyses and rewrote the paper. We are glad that this has clarified many of the questions the reviewer had. We respond to the remaining questions below.

Three big things I would ask the authors to address:

I still think the question of heterogeneity of ea is begging to be addressed here. Especially the intro seems to emphasize it

To address this issue, our manuscript references results from a set of analyses in Okbay et al. 2016 (SI Tables 6.1-6.4)¹ which demonstrate that the effect of the polygenic score for educational attainment (EA) is mediated by several distinct individual characteristics, including higher intelligence, higher openness, and higher conscientiousness. These individual characteristics have imperfect or no genetic correlation with each other (e.g., the COGENT consortium reported genetic correlations of intelligence with openness and conscientiousness of 0.48 [$SE = 0.13$] and 0.10 [$SE = 0.14$], respectively)², suggesting that EA is a genetically heterogeneous trait. We agree that it would be interesting to investigate the genetic heterogeneity of EA further. However, we currently do not have access to additional data that would allow us to conduct meaningful, well-powered analyses over and above what was reported previously^{1,2}. Instead, we extended the description of these analyses and their results in our current version of the paper and emphasized that we are referencing earlier studies that are informative on this issue.

I think the gwis analysis is great, and I don't want to ask the authors for more work, but two things. First, I don't buy the sentence line 375 that the gwis/ldsc results provide evidence of sz heterogeneity at any sort of individual level. If some snps were associated with bip,sz,ea and others are associated only with sz, even while all snps are uniformly associated with any/all subsets of sz patients, would you not get these same results?

The GWIS/LDSC results we refer to here are consistent with the idea that some loci increase SZ liability, BIP and EA while other loci only raise SZ, but not BIP and EA. This implies that SZ patients with a high liability through the first set of loci will have increases BIP and EA liability -- these subjects can reasonably be expected to present differently (i.e with increased EA but also some features of BIP). Vice versa, SZ patients who predominantly carry risk alleles unrelated to EA or BIP could reasonably be expected to have less BIP and EA features. We would argue that these groups of

individuals could be viewed as subsets of SZ patients. However, follow-up work in larger samples would be desirable to identify these sets of patients more clearly from the genetic data.

Second, I just put this out there: I haven't done any real validation analyses of gwis yet, and things in figure 3 like gwis-sz is as similar to bip as it is the sz, and of course gwis-sz,gwis-bip $r_g = -1$, make me wonder if it is doing something biologically interpretable or more just mathematically contrived. If there are some validation analyses you've done that make you feel better about the results, then it might be nice to present them.

The GWIS method³ indeed induces $r_g \approx 0$ between gwis(sz-bip) and bip by construction. The same holds for r_g between gwis(bip-sz) and sz, obviously.

One way to think about GWIS is in the context of structural equation models that allow identifying latent factors that are underlying the observed correlations between GWAS summary statistics of various traits. GWIS subtracts the shared common factor from two sets of GWAS summary statistics and isolates the “unique” genetic components of each trait. An alternative way to get at this would be to model the shared latent factor explicitly and to test if the remaining “unique” components of SZ and BIP would show a similar relationship with EA than what we see with GWIS. Indeed, this is one of the applications that is featured in a new paper⁴ which develops a method to fit structural equation models to GWAS summary statistics. Clearly, the results presented in SI Fig. 7 of that paper⁴ are very similar to our GWIS results. Thus, we have great confidence that our GWIS results are meaningful and correct and we added a reference to the Genomic SEM paper⁴ to our manuscript.

Third big thing, the cautions of the paintor results are fine, but more could be presented (without much more work) that actually might detract less from the impact than these changes. Importantly, the key here is that both concordant and discordant loci need to have at least some evidence of pleiotropy for the conclusions of the paper to hold, not one or the other. Can the authors make any statement to this effect? Possible additions: 1) I would think 80% or even 50% credible sets could be used. What do their sizes and overlaps look like?

Until now, our paper included PAINTOR results with 90% and 65% credible sets. We expanded these analyses and also report PAINTOR results for 80% and 50% credible sets, as the referee suggested. We report these results in **Supplementary Note 7** and **Supplementary Table 3**. Furthermore, we report the results of the broadest set (90%) and the narrowest set (50%) of PAINTOR analyses in Table 1 of the main manuscript. Indeed, the analyses suggest that several of the 21 SNPs identified by our proxy-phenotype analyses are quite likely to have direct pleiotropic effects on SZ and EA.

Specifically, for the broadest credibility set analyses (90%), we found eleven loci with a medium or high credibility to have direct causal effects on both EA and SZ (including one of the novel SNPs, rs7336518). Six of these loci have concordant effects on the two traits (i.e., ++ or --) while five have a discordant effects (i.e., +- or -+, **Table 1** and **Supplementary Note 7**). The narrow credible sets (50%) show that four specific loci (rs7610856, rs320700, rs79210963, and rs7336518) had credibility of more than 15% for the other trait, providing support for the high (rs320700 and rs79210963) and medium (rs7610856 and rs7336518) credibility judgments based on the 90% sets. One of these has a discordant effect (rs7610856) while the others have a concordant effect on SZ and EA.

2) I find myself wanting to look at locuszoom plots in ea and sz for the shared loci; usually it's obvious if peaks might be overlapping.

We included LocusZoom plots (**Supplementary Fig. 3**) that illustrate the three scenarios of interest: (1) a concordant SNP that is likely to have pleiotropic effect (rs79210963) (**Supplementary Fig. 3a**), (2) a discordant SNP that is likely to have pleiotropic effect (rs7610856) (**Supplementary Fig. 3b**), and (3) a SNP with low probability of pleiotropy (rs11694989) (**Supplementary Fig. 3c**). In all three cases, the SNP of interest is the top SNP from the GWAS on EA and the plots illustrate the LD-structure of this SNP in both the EA and SZ GWAS results. The concordant SNP (rs79210963) (**Supplementary Fig. 3a**) is in high LD ($r_{LD}^2 = 0.96$) with the most strongly associated SNP for SZ in this region for which LD information was available (rs112509803). The same holds for the discordant SNP (rs7610856) (**Supplementary Fig. 3b**) which is in high LD ($r_{LD}^2 = 0.98$) with the top hit for SZ in this region (rs62244884). In contrast, the example SNP with low probability for pleiotropy (rs11694989) (**Supplementary Fig. 3c**) is the top hit for EA, but it is clearly not the top hit for SZ in that region (rs6704768). Indeed, the plot shows that the LD between rs11694989 and the top SZ in this region is low ($r_{LD}^2 = 0.27$). The region is also characterized by several short genes in close proximity to each other, suggesting that rs11694989 and the top hit for SZ in this region (rs6704768) could tag different genes.

We discuss these results in **Supplementary Note 7**.

Other things:

To me, if a nature communications paper is a nature or nature genetics paper that's simply not bogged down in endless review, then it might be shorter and more on message. Then, the new loci and pathway/tissues/genes stuff is totally unnecessary, validation in other genetic data and assortative mating could be single sentences here or there, and the polygenic scoring and gwis stuff is what it's all about. Editorially, could maximize the impact of this paper?

We shortened the main text sections on bio annotation (-63 words) and validation (-51 words). We also deleted the section on assortative mating in the main text. Instead, we included a single sentence describing the results of our simulations in the section that reports the proxy-phenotype results (-38 words). We are still reporting all analyses we conducted in the Supplementary Notes for good scientific practice and because some readers may be particularly interested in the sections we shortened in the main paper.

I find the ld aware enrichment section unsatisfying. Overall, my impression is that it's results are the same as the enrichment results with matched snps reported above it, so I don't see why it's not further above or why it's there at all. If depression and related traits are not enriched for sz, then the method lacks power, and the statement of phenotypic specificity is less compelling. How is one snp enriched? Overlap of sz and bip is not new, so is there something new about this result?

The primary purpose of the LD-aware enrichment test we developed is to probe the null hypothesis that enrichment of certain SNPs for various traits (e.g. EA and SZ) is entirely due to LD. Our proxy-phenotype analysis does not specifically control for LD. In particular, the "raw" enrichment reported in **Figure 2b** and in **Supplementary Note 6.4.1** could in principle be due to the LD-structure of the EA lead SNPs that we tested. Specifically, if these EA lead SNPs have stronger LD with other SNPs in the human genome than expected by chance, this could cause the observed enrichment of this set of SNPs on SZ and other traits because higher LD increases the chance these SNPs would "tag" causal SNPs that they are correlated with.

Our LD-aware enrichment calculates the expected test statistic of each SNP, given the heritability of the investigated SNP, GWAS sample size, and the LD-score of the SNP (**Supplementary Note 9**). In principle, this is very similar to the well-known LD-score regression approach⁵. This gives us the “LD-aware” enrichment of each SNP for any GWAS outcome we investigate. When we apply this test to the EA lead-SNPs and their relation to SZ, we clearly reject the null hypothesis that these SNPs are only enriched for association with SZ because of their LD load ($p = 8.43 \times 10^{-13}$ after Bonferroni correction, **Supplementary Table 5.1**). We believe this is an important insight for the interpretation of our data and we clarified this further in the main text.

We note that our LD-aware enrichment test (**Supplementary Figure 5**) does not probe the genetic relationships between depression and SZ per se. Rather, we test if the 132 lead SNPs for EA ($P_{EA} < 10^{-5}$) that are also associated with SZ ($P_{SZ} < 0.05$) show evidence for enrichment with depressive symptoms. In other words, we first sort all SNPs based on whether they are associated with EA. Then we focus on the subset of 132 appropriately independent EA-lead SNPs that is also associated with SZ and conduct the LD-aware enrichment test with them. The genetic correlation between EA and depression is very low ($r_g < 0.001$, $SE = 0.087$)⁶. Thus, finding no LD-aware enrichment of this particular set of SNPs for depressive symptoms is in agreement with expectations based on the genetic correlations among these two phenotypes.

The referee is obviously right that the LD-aware enrichment test is only as good as its statistical power, which is primarily a function of the sample size of the GWAS summary statistics that are being used and the (true) effect sizes of each SNP. We note in this context that the GWAS summary statistics for depressive symptoms that we used were based on a much larger sample ($N = 180,866$) than those for childhood intelligence ($N = 17,989$), although we find clear evidence for LD-aware enrichment with the latter ($p = 6.05 \times 10^{-4}$ after Bonferroni correction, **Supplementary Table 5.1**). Obviously, these results may change as sample sizes for all traits keep growing. We have clarified these issues in the main text.

Line 190- Odds should be or

Thank you, we fixed this here and in all other parts of the main text and the SI.

Lines 270-280- is ea being tested or sz? Are the enrichments for bip, menarche, with sz or with ea? Unclear

This paragraph reports the results of the LD-aware enrichment test across phenotypes for the set of 132 SNPs that are jointly associated with EA ($P_{EA} < 10^{-5}$) and SZ ($P_{SZ} < 0.05$), i.e. the loci that were identified by using EA as a proxy-phenotype for SZ. We clarified this in the main text.

L 309, one per ld block would lead to far fewer than 300000. Ld pruning would leave closer to 100000 in my experience. What are the parameters here? If there are tons of rare snps, perhaps the analysis would improve by excluding them

One would get something closer to 100,000 independent SNPs if only directly genotyped SNPs would be used for the construction of polygenic scores. However, we are using 1000 Genomes imputed data, which provides better coverage than directly genotyped SNPs. Alternatively, one would get a lower number of independent SNPs by penalizing for weak LD between loci even more strongly than we do. However, this would be expected to decrease the predictive power of the scores⁷.

We described all relevant technical details about the construction of our polygenic scores in **Supplementary Note 11.1**. For example, for the SZ score, we used all 8,240,280 SNPs that survived quality control. Next, we applied the clumping procedure in PLINK 1.9 using $r^2 > 0.1$ and 1,000 kb as the clumping parameters and the 1000 Genomes phase 1 version 3 European reference panel to estimate LD among SNPs, eventually leaving a set of 349,357 SNPs ready for profile scoring. In our

experience, this is a typical number of SNPs that one obtains by applying these clumping parameters to 1000 Genomes imputed GWAS results.

We added histograms of the MAF distribution of the SNPs we included in our scores in the SI (**Supplementary Figure 6**). The vast majority of the SNPs we included have $MAF > 1\%$ & $MAF < 99\%$ (80% for EA and 79% for SZ). We clarified this in **Supplementary Note 11.1**.

Furthermore, we conducted robustness checks of our polygenic prediction analyses using scores that excluded SNPs with $99\% < MAF < 1\%$ and $90\% < MAF < 10\%$, respectively. This resulted in scores with $\approx 300,000$ and $\approx 100,000$ LD-pruned SNPs, respectively. We report these additional robustness checks in **Supplementary Notes 11.1.2, 12.3** and **Supplementary Table 8.7**. Overall, the results we obtain are very similar to our main model specification.

Page 9. Is the f test really appropriate here? Should aic or bic be used, since the models are not hierarchical/nested? Honestly this is probably not very important.

The total PGS, S , can be written as the sum of two scores, say T and U (where $S = T + U$, by virtue of the fact that the set of SNPs is partitioned into two subsets, with one subset used for the construction of T and the other for the construction of U). Hence, the regression of an outcome, Y , on T and U is identical to a regression of Y on S and U in the following sense: the subspace spanned by T and U is identical to the subspace spanned by S and U . Ergo, the regression of Y on T and U is nested with respect to the regression of Y on S , in terms of the subspaces being spanned by respective sets of regressors, enabling us to formally test the hypothesis of genetic homogeneity with an F -test.

We clarified this in **Supplementary Note 1.3.2**.

Reviewer #3 (Remarks to the Author):

Paper is improved, and additional analyses have been carried out. I would suggest that the data supports the possibility that bipolar -related SNPs mediate the relationship observed between EA and SZ, possibly through higher energy/drive. This possible mediating relationship of Bipolar SNPs makes things less counter-intuitive, and needs to be mentioned more strongly in the discussion and/or abstract.

We thank the reviewer for sharing this interesting thought with us. We now mention this in the discussion section of our manuscript, while trying not to be too speculative:

“Instead, our results are most consistent with the idea that EA and SZ are both genetically heterogeneous traits that aggregate over various subphenotypes or symptoms with non-identical genetic architectures. Specifically, our results suggest that current SZ diagnoses aggregate over at least two disease subtypes: One part resembles BIP and high IQ (possibly associated with *Concordant* SNPs), where better cognition may also be genetically linked to other BIP features such as higher energy and drive, while the other part is a cognitive disorder that is independent of BIP (possibly influenced by *Discordant* SNPs). This latter subtype bears similarity with Kraepelin’s description of dementia praecox⁸. Overall, our pattern of results resonates with the idea that cognitive deficits in early life may be an important differentiating factor between patients with BIP versus SZ psychosis.”

References

1. Okbay, A. *et al.* Genome-wide association study identifies 74 loci associated with educational attainment. *Nature* **533**, 539–542 (2016).
2. Trampush, J. W. *et al.* GWAS meta-analysis reveals novel loci and genetic correlates for general cognitive function: a report from the COGENT consortium. *Mol. Psychiatry* **22**, 336–345 (2017).
3. Nieuwboer, H. A., Pool, R., Dolan, C. V., Boomsma, D. I. & Nivard, M. G. GWIS: Genome-wide inferred statistics for functions of multiple phenotypes. *Am. J. Hum. Genet.* **99**, 917–927 (2016).
4. Grotzinger, A. D. *et al.* Genomic SEM provides insights into the multivariate genetic architecture of complex traits. *bioRxiv* (Cold Spring Harbor Laboratory, 2018). doi:10.1101/305029
5. Bulik-Sullivan, B. K. *et al.* LD Score regression distinguishes confounding from polygenicity in genome-wide association studies. *Nat. Genet.* **47**, 291–295 (2015).
6. Bulik-Sullivan, B. K. *et al.* An atlas of genetic correlations across human diseases and traits. *Nat. Genet.* **47**, 1236–1241 (2015).
7. Vilhjálmsson, B. J. *et al.* Modeling linkage disequilibrium increases accuracy of polygenic risk scores. *Am. J. Hum. Genet.* **97**, 576–592 (2015).
8. Kraepelin, E. *Psychiatrie: Ein Lehrbuch für Studierende und Ärzte.* (Verlag von Johann Ambrosius Barth, 1893).